# SPD: Synergy Pattern Diversifying Oriented Unsupervised Multi-agent Reinforcement Learning

**Yuhang Jiang,**\* **Jianzhun Shao,**\* **Shuncheng He, Hongchang Zhang, Xiangyang Ji**
Department of Automation
Tsinghua University, Beijing, China
{jiangyh19, sjz18, hesc16, hc-zhang19}@mails.tsinghua.edu.cn
xyji@tsinghua.edu.cn

## Abstract

Reinforcement learning typically relies heavily on a well-designed reward signal, which gets more challenging in cooperative multi-agent reinforcement learning. Alternatively, unsupervised reinforcement learning (URL) has delivered on its promise in the recent past to learn useful skills and explore the environment without external supervised signals. These approaches mainly aimed for the single agent to reach distinguishable states, insufficient for multi-agent systems due to that each agent interacts with not only the environment, but also the other agents. We propose *Synergy Pattern Diversifying Oriented Unsupervised Multi-agent Reinforcement Learning* (SPD) to learn generic coordination policies for agents with no extrinsic reward. Specifically, we devise the *Synergy Pattern Graph* (SPG), a graph depicting the relationships of agents at each time step. Furthermore, we propose an episode-wise divergence measurement to approximate the *discrepancy of synergy patterns*. To overcome the challenge of sparse return, we decompose the discrepancy of synergy patterns to per-time-step pseudo-reward. Empirically, we show the capacity of SPD to acquire meaningful coordination policies, such as maintaining specific formations in Multi-Agent Particle Environment and pass-and-shoot in Google Research Football. Furthermore, we demonstrate that the same instructive pretrained policy's parameters can serve as a good initialization for a series of downstream tasks' policies, achieving higher data efficiency and outperforming state-of-the-art approaches in Google Research Football.

## 1 Introduction

Deep reinforcement learning (RL) has exhibited excellent capabilities in tackling complex tasks, including Go [40, 36], poker [27] and simulated robotic tasks [37, 13]. However, one of the key challenges in deploying RL algorithms is to devise an elaborate reward signal for eliciting desired behaviour of agents [14]. Due to the sharing of the team reward, the situation gets more complicated for cooperative multi-agent reinforcement learning (MARL), especially in real-world applications such as coordination of robot swarms [16] and autonomous vehicle teams [52], where verifying the effectiveness of the designed reward can be prohibitively costly.

As for the single agent, unsupervised learning has been incorporated into RL to acquire diverse *skills* for the agent without extrinsic reward from the environment, and this scenario is known as unsupervised reinforcement learning (URL). Recent advances in URL [10, 15, 20, 31, 38] have demonstrated the potential of the agent to explore the environment and learn useful skills with no supervision. Moreover, when endowed with the task-related extrinsic reward in downstream tasks, these skills exhibit significant generalizability, leading to efficient adaptation. An important

---

\*Equal contribution.

36th Conference on Neural Information Processing Systems (NeurIPS 2022).

hypothesis explaining the phenomenon is made by DIAYN [10] that the higher the coverage of learned skills over the set of possible behaviour, the greater the probability of gaining effective skills.

Above mentioned URL approaches mostly depict the diversity of skills by the diversity of visited states, however, insufficient for MARL due to the neglect of interactions with other agents. Intuitively, this coordination "skill" in MARL corresponding to the skill in single-agent URL should be taken into consideration as well. We term the coordination "skill" as the *Synergy Pattern*, which is the perennial coordinated behaviour of agents induced by interactions of joint policies and the environment. In retrospect, the coordination of agents are usually modeled by *graphs*, such as coordination graphs [12, 4, 50] and graph neural networks [17, 21]. Nevertheless, these graph-based approaches are mainly tasked for communication and deploying the graphs directly to unsupervised MARL can be challenging.

We diversify the synergy patterns of agents by bridging the graphs and unsupervised MARL. Specifically, we devise the *Synergy Pattern Graph* (SPG), in which there is a vertex for each agent and the weight of the edge is derived from the function depicting the relations of agents, as the per-time-step instance of the synergy pattern. Our method *Synergy Pattern Diversifying Oriented Unsupervised Multi-agent Reinforcement Learning* (SPD) focuses on obtaining diverse synergy patterns while following the same hypothesis in URL. To this end, we approximate the *Discrepancy of Synergy Patterns* by quantifying the divergence between two SPG batches, which are sampled from the distributions of different synergy patterns respectively. The first step is to measure the discrepancy between two SPG elements. This problem is essentially to measure the similarity of graphs, which has been well-studied in the field of graph matching[39, 26, 32, 9], and one of the promising approaches is the Gromov-Wasserstein discrepancy [6, 51]. With the adoption of the Gromov-Wasserstein discrepancy as the pair-wise measurement, we then formulate the discrepancy of synergy patterns as an episode-wise measurement and deploy it as the pseudo episode return for unsupervised MARL.

Our contributions for SPD, a general framework for unsupervised MARL, are summarized as follows: i) we devise SPG with a general formulation as the instance to depict the synergy patterns of agents; ii) we propose an episode-wise divergence measurement to estimate the discrepancy of synergy patterns; iii) we derive the formulation for the decomposition of the pseudo episode return to alleviate the problem of the sparse return; iv) The empirical results demonstrate that SPD achieves better performance in MARL compared to conventional URL approaches and shows great potentials to learn synergy patterns with generalizability for downstream tasks.

## 2 Related works

**Multi-agent reinforcement learning (MARL)** has shown great potential in solving multi-agent cooperation tasks. A popular paradigm in MARL is the centralized training with decentralized execution (CTDE) that combines the advantage of independent Q-learning [44] and joint action learning [7]. Centralized training makes agents cooperate better while decentralized execution benefits the flexible deployment capability. We mainly focus on the value-based CTDE methods. Following the CTDE paradigm, a series of works concentrate on distributing the team reward to all agents by value function factorization [43, 34], and deriving the Individual-global-max principle for policy optimality analysis [42, 46, 35]. Another bunch of works introduce roles or skills to agents for better team cooperation [24, 48, 5]. However, these methods usually suffer from poor generalization ability and have considerable performance drop when transferred to unseen scenarios. Wang et al. [49] reports agents' generalization by splitting the action space, but the method relies heavily on the handcrafted hyper-parameter. The concept of agents' forming a graph is also widely used to represent agents' cooperation in MARL, including the coordination graph [4, 50] and graph neural networks [17, 21]. But these methods are usually designed for communication, which brings extra costs for decentralized execution. In contrast to these methods, our proposed method uses the synergy graph to train team skills without rewards from the environment, which brings the agents strong generalization ability in downstream tasks, and takes no extra cost in execution.

**Unsupervised reinforcement learning (URL)** has been particularly intriguing in the context of RL, consider the high cost of real-world rollouts with delicately designed reward signal. This paper is concerned about unsupervised behaviour learning for deep RL that requires learning with only the task-agnostic intrinsic reward. Early works [11, 1] mainly focuses on controlling the visited states via regulating a latent variable $Z$, where the mutual information is maximized with visited

states. DIAYN [10] demonstrates the exceptional performance of URL by maximizing $I(Z; S)$ while simultaneously minimizing $I(Z; A|S)$. The recent studies have also shown interests on adapting pre-trained skills on MDPs from aspects of the initial state [3] and the dynamics model [38]. Besides, WURL [15] adopts the idea of optimal transport and increases the discrepancy of learned skills by enlarging the distance of visited state distributions estimated by Wasserstein distance. These approaches aiming for achieving easily distinguishable states, however, they are deficient to solve the growing size of joint observation space in multi-agent systems. By contrast, our method devise a graph to depict the relative relations between agents which alleviate the growing size challenge. There are also works encouraging diversity for effective exploration [19, 23, 30, 53], while they mostly enhance single policy's diversity or the inter-agent diversity and do not get rid of the dependence of the task-related reward. We follow a similar hypothesis to that in Eysenbach et al. [10] to learn diverse synergy patterns for gaining useful coordination policies.

## 3 Preliminaries

### 3.1 Notation

We consider a Dec-POMDP [29] for the fully cooperative multi-agent task with $n$ agents, defined by a tuple $\langle \mathcal{S}, \mathcal{A}, \mathcal{P}, \mathcal{R}, \mathcal{O}, \mathcal{I}, \Omega, n, \gamma \rangle$, where $s \in \mathcal{S}$ and $a^i \in \mathcal{A}$ denote the true state of the environment and the discrete action chosen by agent $i \in \mathcal{I} \equiv \{1, \ldots, n\}$, respectively. $\gamma \in [0, 1)$ is the discount factor. The dynamics of the environment can be formulated as the state transition function $\mathcal{P}(s'|s, \boldsymbol{a}) : \mathcal{S} \times \boldsymbol{A} \times \mathcal{S} \to [0, 1]$ and the reward function $\mathcal{R}(s, \boldsymbol{a}) : \mathcal{S} \times \boldsymbol{A} \to \mathbb{R}$ shared by all agents, where $\boldsymbol{a} \in \boldsymbol{A} \equiv \mathcal{A}^n$ denotes the joint action. Each agent $i$ only has access to its own observation $o^i \in \mathcal{O}$ derived from the observation function $\Omega(s, i) : \mathcal{S} \times \mathcal{I} \to \mathcal{O}$, due to the *partial observability*.

At each time step $t$, agent $i$ resolves to execute the action $a^i$ according to a stochastic policy $\pi^i(a^i|\tau^i) : \mathcal{T} \times \mathcal{A} \to [0, 1]$, where $\tau^i \in \mathcal{T} \equiv (\mathcal{O} \times \mathcal{A})^*$ denotes the action-observation history of the agent $i$. The goal of MARL is to find an optimal joint policy $\boldsymbol{\pi}^* \equiv \prod_{i=1}^n \pi^{i*}$ satisfying the condition: $Q^{\boldsymbol{\pi}^*}(s, \boldsymbol{a}) \geq Q^{\boldsymbol{\pi}}(s, \boldsymbol{a}), \forall \boldsymbol{\pi}$ and $(s, \boldsymbol{a}) \in \mathcal{S} \times \boldsymbol{A}$, where $Q^{\boldsymbol{\pi}}(s_t, \boldsymbol{a}_t) = \mathbb{E}_{s_{t+1:\infty}, \boldsymbol{a}_{t+1:\infty}} \left[ \sum_{i=0}^\infty \gamma^i r_{t+i}|s_t, \boldsymbol{a}_t \right]$ is the joint action-value function induced by joint policy $\boldsymbol{\pi}$.

We adopt the paradigm of centralized training with decentralized execution (CTDE). During training the true state $s$ and the joint action-observation history $\boldsymbol{\tau} \equiv \{\tau^i\}_{i=1}^n$ are accessible to the learning algorithm, while each agent $i$ selects an action based on the policy conditioning only on its own action-observation history $\tau^i$ during testing (execution).

### 3.2 Optimal transport and Gromov-Wasserstein discrepancy

Optimal transport (OT) is a general problem of moving one distribution of mass to another as efficiently as possible. Likewise, the distinction of two given graphs can be regarded as an optimal transport problem to find the correspondence between their vertices and edges. Previous work [6] proposed *Gromov-Wasserstein Discrepancy*, which is extended from Gromov-Wasserstein distance [25] as a measure for the divergence of the graphs.

**Definition 3.1.** *[6, 51] (Gromov-Wasserstein discrepancy) Denote a measure graph as $G(\mathcal{V}, \boldsymbol{\mathcal{W}}, \boldsymbol{\mu})$, where $\mathcal{V} \equiv \{v_i\}_{i=1}^{|\mathcal{V}|}$ is the set of vertices, $\omega_{ij} \in \boldsymbol{\mathcal{W}}$ is the weight of the undirected edge $\{i, j\}$ between vertices $v_i, v_j \in \mathcal{V}$, and $\boldsymbol{\mu} = [\mu_i] \in \Sigma^{\mathcal{V}}$ is a Borel probability measure defined on $\mathcal{V}$. $\forall p \in [1, \infty]$ and each $G_\nu, G_\varsigma \in \mathcal{G}$, where $\mathcal{G}$ denotes the collection of measure graphs, the Gromov-Wasserstein discrepancy between $G_\nu$ and $G_\varsigma$ can be derived as*

$$d_{gw}(G_\nu, G_\varsigma) := \min_{\boldsymbol{T} \in \Pi(\boldsymbol{\mu}_\nu, \boldsymbol{\mu}_\varsigma)} \left( \sum_{i,j \in \mathcal{V}_\nu} \sum_{i',j' \in \mathcal{V}_\varsigma} \left| \omega_{ij}^\nu - \omega_{i'j'}^\varsigma \right|^p T_{ii'} T_{jj'} \right)^{\frac{1}{p}}, \tag{1}$$

*where* $\Pi(\boldsymbol{\mu}_\nu, \boldsymbol{\mu}_\varsigma) = \left\{ \boldsymbol{T} \geq \boldsymbol{0} \mid \boldsymbol{T} \boldsymbol{1}_{|\mathcal{V}_\varsigma|} = \boldsymbol{\mu}_\nu, \boldsymbol{T}^\top \boldsymbol{1}_{|\mathcal{V}_\nu|} = \boldsymbol{\mu}_\varsigma \right\}.$

In general, the Gromov-Wasserstein discrepancy measures the divergence between two graphs by comparing the differences of the edges' relationship in one graph to that in the other. We refer the reader to Mémoli [25], Chowdhury & Mémoli [6], Peyré et al. [33] for mathematical foundations.

# 4 Methodology

This section is organized as the following: In Sec. 4.1, we delineate the concept of *Synergy Pattern* and propose a novel structure of graph as the instance of it, namely *Synergy Pattern Graph.* We then carefully derive the formulation for *Discrepancy of Synergy Patterns* and make an approximation of it for convenience in practice in Sec. 4.2. In Sec. 4.3, our unsupervised framework SPD is depicted in detail and the mechanism for the decomposition of the pseudo reward is explained. Sec. 4.4 provides the implementation details for solving the optimization problem.

## 4.1 Synergy pattern

With the awareness of interactions between agents' actions, we describe the perennial coordinated behaviour of agents by the *Synergy Pattern*. Previous methods [12, 4, 17, 21] utilize the structure of *graph* to depict it, whereas most of them focus on the local relationship between agent pairs towards a specific task. To get rid of the extrinsic reward signal, we propose *Synergy Pattern Graph* (SPG), denoted as $G^{sp}(\mathcal{V}, \mathcal{W}, \boldsymbol{\mu}, \zeta)$, where $v_i \in \mathcal{V}$ is the vertex for agent $i \in \mathcal{I}$ and the weight $\omega_{ij} \in \mathcal{W}$ of edge $\{i, j\}$ is derived from a *synergy pattern function* which could depict agents' relative relations:

$$\zeta(\tau^i, \tau^j) : \mathcal{T} \times \mathcal{T} \to \mathbb{R}_0^+$$

The synergy pattern function follows a general formulation as any function with range $\mathbb{R}_0^+$, for instance, the information-theoretic influence [47]: $\zeta(\tau^i, \tau^j) := \mathrm{MI}_{j|i}^{\boldsymbol{\pi}}(o'^j; o^i, a^i | o^j, a^j)$ or the communication messages [4]. It can depend on more than merely the observations and joint actions as well, whereas we omit this for simplicity. At each time step, SPG varies depending on the history of the multi-agent system rather than being predefined before training. We will use $G_t$ to denote $G^{sp}(\mathcal{V}, \mathcal{W}, \boldsymbol{\mu}, \zeta)$ at time step $t$ in the following part of this paper for ease of notation.

Accordingly, the synergy pattern can be described by the distribution of SPG obtained by deploying a joint policy $\boldsymbol{\pi}$ in the environment: $\mathbb{P}(G^{\boldsymbol{\pi}} | \boldsymbol{\pi})$ with domain $\Lambda$, where $\Lambda$ is the collection of all SPG.

## 4.2 Discrepancy of synergy patterns

We use the discrepancy between two distributions of SPG: $\mathbb{P}(G^{\boldsymbol{\pi}_1} | \boldsymbol{\pi_1})$ and $\mathbb{P}(G^{\boldsymbol{\pi}_2} | \boldsymbol{\pi_2})$ to denote the discrepancy between two synergy patterns. This process can be decomposed into two phases: i) measuring the distance between two samples from each distribution respectively; ii) extending the metric for a pair of samples to that for distributions. The former one is essentially to estimate the discrepancy between graphs. We adopt the Gromov-Wasserstein discrepancy of graphs, as mentioned in section 3.2, to measure this distance. Besides, we solve the latter one by applying optimal transport on SPG distributions.

This whole process leads to the following definition:

**Definition 4.1.** *(Discrepancy of Synergy Patterns) Let $p_1 = \mathbb{P}(G^{\boldsymbol{\pi}_1} | \boldsymbol{\pi_1})$ and $p_2 = \mathbb{P}(G^{\boldsymbol{\pi}_2} | \boldsymbol{\pi_2})$ be two distributions of SPG induced by joint polices $\boldsymbol{\pi}_1$ and $\boldsymbol{\pi}_2$, respectively. Denote an joint distribution $\varrho \in \Gamma[p_1, p_2]$ as a bijective plan transporting probability mass from $p_1$ to $p_2$, where $\Gamma[p_1, p_2]$ is the set of all distributions on product space $\Lambda \times \Lambda$ with $p_1$ and $p_2$ as their marginal probabilities. Then the discrepancy of synergy patterns is defined as*

$$d_{sp}(p_1, p_2) = \inf_{\varrho \in \Gamma[p_1, p_2]} \int d_{gw}(G^{\boldsymbol{\pi}_1}, G^{\boldsymbol{\pi}_2}) \, \mathrm{d}\varrho \qquad (2)$$

where $d_{gw}$ is the Gromov-Wasserstein discrepancy according to Eq. (1). Besides, we show that $d_{sp}$ is a legitimate pseudometric on $\Lambda$ and the proof is deferred to Appendix A.

However, solving Eq. (2) for the discrepancy of synergy patterns is almost intractable. In practice, we collect two SPG batches to approximate the discrepancy of synergy patterns. Here, we give the definition for this approximation:

**Definition 4.2.** *(Proximal Discrepancy of Synergy Patterns) Denote $\bar{\lambda}^{\boldsymbol{\pi}_1} = \{G_t^{\boldsymbol{\pi}_1}\}_{t=0}^{B_1} \sim \mathbb{P}(G^{\boldsymbol{\pi}_1} | \boldsymbol{\pi_1})$ and $\bar{\lambda}^{\boldsymbol{\pi}_2} = \{G_t^{\boldsymbol{\pi}_2}\}_{t=0}^{B_2} \sim \mathbb{P}(G^{\boldsymbol{\pi}_2} | \boldsymbol{\pi_2})$ sampled from the distributions of SPG, where $B_1, B_2 \in \mathbb{N}^+$. Assume $B_1 \leq B_2$ by symmetry. Let $D_{B_1 \times B_2} = \left[ d_{gw}(G_{t_1}^{\boldsymbol{\pi}_1}, G_{t_2}^{\boldsymbol{\pi}_2}) \right], \forall 0 \leq t_1 \leq B_1, 0 \leq t_2 \leq B_2$*

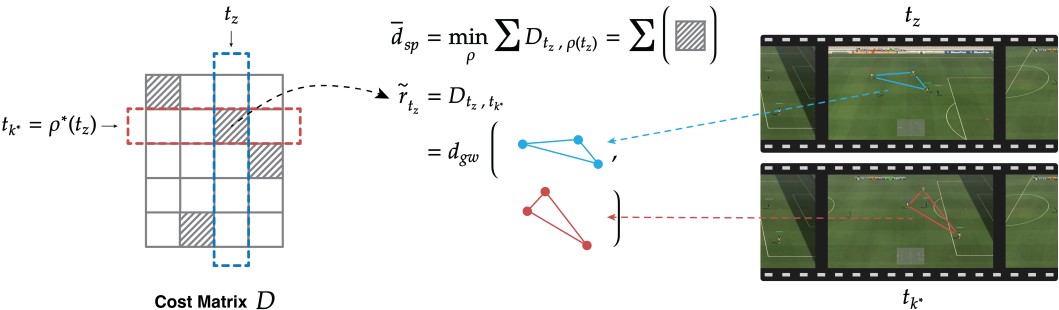

Figure 1: Illustration for the process of obtaining proximal discrepancy of synergy patterns $\bar{d}_{sp}$ (left) and the process of computing pseudo reward (right).

*be the cost matrix. Suppose that an injective function $\rho : \{1, \ldots, B_1\} \to \{1, \ldots, B_2\}$ denotes an matching function. By minimizing the total cost, an approximation for the discrepancy of synergy patterns is:*

$$\bar{d}_{sp}(\bar{\lambda}^{\boldsymbol{\pi}_1}, \bar{\lambda}^{\boldsymbol{\pi}_2}) = \min_{\rho} \sum_{t=0}^{B_1} D_{t,\rho(t)} \tag{3}$$

It is vital to point out that the SPG from $\bar{\lambda}^{\boldsymbol{\pi}_2}$ corresponding to $G_t^{\boldsymbol{\pi}_1}$ may not be $G_t^{\boldsymbol{\pi}_2}$ since the relations of agents may be analogous at different time steps. Hence, we search the whole $\bar{\lambda}^{\boldsymbol{\pi}_2}$ for $G_{\rho(t)}^{\boldsymbol{\pi}_2}$ that is relatively adjacent to $G_t^{\boldsymbol{\pi}_1}$ based on the Gromov-Wasserstein discrepancy.

### 4.3 Synergy pattern diversifying oriented unsupervised multi-agent reinforcement learning

Based on the discrepancy of synergy patterns, we can now delineate the framework for SPD. Suppose that $Z \in \mathbb{N}^+$ denotes the number of different synergy patterns. Our goal is to learn a set of joint policies $\{\boldsymbol{\pi}_z\}_{z=1}^Z$ with diverse synergy patterns to explore the environment sufficiently when there is no extrinsic reward. To increase the divergence among synergy patterns, we formulate the pseudo episode return for the episode data $\boldsymbol{\tau}_z$ produced by the joint policy $\boldsymbol{\pi}_z$:

$$\tilde{R}_z(\boldsymbol{\tau}_z, \{\boldsymbol{\tau}_k\}) = \min_{k \neq z, 1 \leq k \leq Z} \bar{d}_{sp}(\bar{\lambda}^{\boldsymbol{\pi}_z}, \bar{\lambda}^{\boldsymbol{\pi}_k}), \tag{4}$$

where $1 \leq z \leq Z$, and $\bar{\lambda}^{\boldsymbol{\pi}_z}$ is the SPG set derived from $\boldsymbol{\tau}_z$. Besides, we obtain $\bar{\lambda}^{\boldsymbol{\pi}_k}$ by sampling a batch of data from the replay buffer of $\boldsymbol{\pi}_k$ and ensure $\bar{\lambda}^{\boldsymbol{\pi}_k}$ have more elements: $|\bar{\lambda}^{\boldsymbol{\pi}_k}| \geq |\bar{\lambda}^{\boldsymbol{\pi}_z}|$.

Our method is to maximize the expected return: $\max_{\boldsymbol{\pi}_z} \mathbb{E}[\tilde{R}_z(\boldsymbol{\tau}_z, \{\boldsymbol{\tau}_k\})]$, resulting in the maximization of each discrepancy of one synergy pattern to its nearest counterpart. However, the pseudo episode return in Eq. (4) depending on the whole episode data $\boldsymbol{\tau}_z$ is sparse since it only derives a single final reward for one episode. This imposes the trouble for MARL algorithms to learn an accurate action-value function $Q^{\boldsymbol{\pi}_z}(s, \boldsymbol{a})$ and tends to impair the performance of value-based MARL algorithms. To bypass this predicament, intuitively, the pseudo episode return $\tilde{R}_z(\boldsymbol{\tau}_z, \{\boldsymbol{\tau}_k\})$ can be decomposed into pseudo rewards for every time step according to the optimal matching function $\rho^*$. Concretely, the pseudo reward at time step $t$ can be formulated as:

$$\tilde{r}_t(\boldsymbol{\tau}_z, \{\boldsymbol{\tau}_k\}) = d_{gw}(G_t^{\boldsymbol{\pi}_z}, G_{\rho^*(t)}^{\boldsymbol{\pi}_{k^*}}), \quad G_t^{\boldsymbol{\pi}_z} \in \bar{\lambda}^{\boldsymbol{\pi}_z}, G_{\rho^*(t)}^{\boldsymbol{\pi}_{k^*}} \in \bar{\lambda}^{\boldsymbol{\pi}_{k^*}}, \tag{5}$$

where $\rho^*$ is the optimal solution to $\bar{d}_{sp}(\bar{\lambda}^{\boldsymbol{\pi}_z}, \bar{\lambda}^{\boldsymbol{\pi}_k})$ by Eq. (3), and $k^*$ denotes the id of the synergy pattern with minimal discrepancy to $\bar{\lambda}^{\boldsymbol{\pi}_z}$ according to Eq. (4). In other word, the pseudo reward is the Gromov-Wasserstein discrepancy of the SPG at current time step and the SPG from the counterpart of another synergy pattern.

We discriminate the concepts we proposed in Appendix D for a clearer understanding. The visualization for the pseudo reward derivation is shown in Fig. 1.

### 4.4 Implementation

We implement SPD with QMIX [34], learning a set of joint policies that maximize the discrepancy of synergy patterns to each other. QMIX is designed to have a mixing network structure, learning a global action-value function $Q := \{Q_{tot}|Q_{tot}(\boldsymbol{\tau}, \boldsymbol{a}) = f_s\left(Q^1(\tau^1, a^1), \ldots, Q^n(\tau^n, a^n)\right), \frac{\partial f_s}{\partial Q^i} \geq 0, Q^i(\cdot, \cdot) \in \mathbb{R}\}$ which satisfies the IGM principle [42], to accomplish the cooperative task for agents.

During unsupervised learning, we select $1 \leq z \leq Z$ for the synergy pattern at the start of each episode randomly or in turn, and interact with the environment according to the joint policy $\boldsymbol{\pi}_z$ throughout the episode. In practice, the Euclidean distance of $o^i$ and $o^j$ is adopted as the weight $\omega_{ij}$ of edge $\{i, j\}$ to build the SPG, i.e., the formulation of synergy pattern function is $\zeta(\tau^i, \tau^j) = \|o^i - o^j\|_2$. We will show in Sec. 5 that, even with such a synergy pattern function to grab the simple information of relationships of agents, SPD can still learn useful synergy patterns such as pass-and-shoot in Google football Research and accelerate the learning procedure when endowed with the task-related reward.

Since solving for the Gromov-Wasserstein discrepancy between two measure graphs in Eq. (1) is a computationally costly process, we adopt [51]'s Regularized Proximal Gradient method to estimate it. We set $p = 2$ in Eq. (1) in our experiments. As for the probability measure, we set $\boldsymbol{\mu} = \frac{1}{n}\mathbf{1}_n$ since the agents are assumed to have equal contributions, where $n$ is the number of agents or vertices in synergy patterns and $\mathbf{1}_n$ is a size $n$ vector of ones. In the $m$-th iteration, the current optimal transport $\boldsymbol{T}^{(m)}$ can be updated by

$$
\begin{aligned}
\boldsymbol{T}^{(m+1)} &= \underset{\boldsymbol{T} \in \Pi(\boldsymbol{\mu}_\nu, \boldsymbol{\mu}_\varsigma)}{\operatorname{argmin}} \sum_{i,j \in \mathcal{V}_\nu} \sum_{i',j' \in \mathcal{V}_\varsigma} \left|\omega_{ij}^\nu - \omega_{i'j'}^\varsigma\right|^2 T_{ii'}^{(m)} T_{jj'}^{(m)} + \alpha \operatorname{KL}(\boldsymbol{T}\|\boldsymbol{T}^{(m)}) \\
&= \underset{\boldsymbol{T} \in \Pi(\boldsymbol{\mu}_\nu, \boldsymbol{\mu}_\varsigma)}{\operatorname{argmin}} \langle \boldsymbol{L}(\boldsymbol{\mathcal{W}}_\nu, \boldsymbol{\mathcal{W}}_\varsigma, \boldsymbol{T}^{(n)}), \boldsymbol{T}\rangle + \alpha \operatorname{KL}(\boldsymbol{T}\|\boldsymbol{T}^{(n)}),
\end{aligned}
\tag{6}
$$

where $\boldsymbol{L}(\boldsymbol{\mathcal{W}}_\nu, \boldsymbol{\mathcal{W}}_\varsigma, \boldsymbol{T}^{(n)}) = \boldsymbol{\mathcal{W}}_\nu \boldsymbol{\mu}_\nu \mathbf{1}_{|\mathcal{V}_\varsigma|}^\top + \mathbf{1}_{|\mathcal{V}_\nu|} \boldsymbol{\mu}_\varsigma^\top \boldsymbol{\mathcal{W}}_\varsigma^\top - 2\boldsymbol{\mathcal{W}}_\nu \boldsymbol{T} \boldsymbol{\mathcal{W}}_\varsigma^\top$, and Kullback-Leibler (KL) divergence is added as the proximal term. $\langle \cdot, \cdot \rangle$ denotes the inner product of two matrices and $\alpha$ is a hyper-parameter to balance two items. The Eq. (6) can then be solved with nearly-linear convergence [2] by the Sinkhorn-Knopp algorithm [41, 8]. As soon as $\boldsymbol{T}^{(m)}$ converges to a stable optimal transport $\hat{\boldsymbol{T}}$, the estimation of the Gromov-Wasserstein discrepancy can be calculated by plugging $\hat{\boldsymbol{T}}$ into Eq. (1).

To solve Eq. (3), we utilize Kuhn–Munkres algorithm [28], which could solve such an assignment problem in polynomial time, to obtain the optimal matching funciton $\rho^*$. Therefore, the complete formula for pseudo reward at time step $t$ is as below:

$$
\tilde{r}_t(\boldsymbol{\tau}_z, \{\boldsymbol{\tau}_k\}) = d_{gw}(G_t^{\boldsymbol{\pi}_z}, G_{\rho^*(t)}^{\boldsymbol{\pi}_{k^*}}) = \left( \sum_{i,j \in \mathcal{V}_t^{\boldsymbol{\pi}_z}} \sum_{i',j' \in \mathcal{V}_{\rho^*(t)}^{\boldsymbol{\pi}_{k^*}}} \left|\omega_{t,(ij)}^{\boldsymbol{\pi}_z} - \omega_{\rho^*(t),(i'j')}^{\boldsymbol{\pi}_{k^*}}\right|^2 \hat{T}_{ii'} \hat{T}_{jj'} \right)^{\frac{1}{2}},
\tag{7}
$$

where $G_t^{\boldsymbol{\pi}_z} \in \bar{\lambda}^{\boldsymbol{\pi}_z}, G_{\rho^*(t)}^{\boldsymbol{\pi}_{k^*}} \in \bar{\lambda}^{\boldsymbol{\pi}_{k^*}}$, and $k^*$ denotes the id of the synergy pattern with minimal discrepancy to $\bar{\lambda}^{\boldsymbol{\pi}_z}$ according to Eq. (4). The whole framework is summarized in Algorithm 1.

## 5 Experiments

In this section, we first compare properties of SPD with that of other URL baselines on Multi-agent Particle Environment (MPE) [22] in Sec. 5.1. And in Sec. 5.2, we evaluate the generalization ability of our method on downstream tasks on Google Research Football (GRF) [18]. Our code is available at `https://github.com/thu-rllab/SPD`.

### 5.1 Synergy pattern diversifying

We demonstrate that our method SPD is capable of capturing more inherent properties of multi-agent systems such as the impacts from other agents, comparing to conventional URL algorithms. In order

---
**Algorithm 1** SPD
---

**Input**: $Z, t_{\text{start}}$. Empty graph buffers $\{\mathcal{D}_k\}_{k=1}^Z$. Empty replay buffers $\{\mathcal{D}_k^{RL}\}_{k=1}^Z$. The synergy pattern function $\zeta$ as described in section 4.1. Maximal time steps for each episode $t_{\text{max}}$.
**Parameter**: joint policy networks $\{\boldsymbol{\pi}_{\theta_k}\}_{k=1}^Z$

1: Initialize $t_{\text{env}} = 0, \{\theta_k\}_{k=1}^Z$.
2: **for** *each episode* **do**
3:     Select $z$ randomly or in turn. $\boldsymbol{\tau}_z \leftarrow \{\}, \bar{\lambda}^{\boldsymbol{\pi}_z} \leftarrow \{\}$;
4:     **while** not *done* **do**
5:         Interact with the environment by $\boldsymbol{\pi}_z$ to get the tuple $(s, \boldsymbol{a}, s')$;
6:         $\boldsymbol{\tau}_z \leftarrow \boldsymbol{\tau}_z \cup \{(s, \boldsymbol{a}, s')\}$.
7:         Build SPG $G$ with $\zeta(\boldsymbol{\tau}_z)$;
8:         $\bar{\lambda}^{\boldsymbol{\pi}_z} \leftarrow \bar{\lambda}^{\boldsymbol{\pi}_z} \cup \{G\}, \mathcal{D}_z \leftarrow \mathcal{D}_z \cup \{G\}$;
9:         $t_{\text{env}} \leftarrow t_{\text{env}} + 1$;
10:     **end while**
11:     **if** $t_{\text{env}} \geq t_{\text{start}}$ **then**
12:         **for** *each* $k \neq z$ **do**
13:             $\bar{\lambda}^{\boldsymbol{\pi}_k} \sim \mathcal{D}_k$.
14:             # via Sinkhorn $-$ Knopp algorithm
15:             Obtain $d_{gw}(G^{\boldsymbol{\pi}_z}, G^{\boldsymbol{\pi}_k})$ according to Eq. (6) as well as Eq. (1);
16:             # via Kuhn $-$ Munkres algorithm
17:             Obtain $\bar{d}_{sp}^k(\bar{\lambda}^{\boldsymbol{\pi}_z}, \bar{\lambda}^{\boldsymbol{\pi}_k})$ by solving Eq. (3) for $\rho^*$;
18:         **end for**
19:         $k^* \leftarrow \arg\min_k \bar{d}_{sp}^k$;
20:         Calculate $\tilde{r}$ by plugging $k^*, \rho^{k^*}$ into Eq. (7);
21:         $\mathcal{D}_z^{RL} \leftarrow \mathcal{D}_z^{RL} \cup \{\boldsymbol{\tau}_z, \{\tilde{r}\}_{t=1}^{t_{\text{max}}}\}$;
22:         Use an MARL algorithm to train $\theta_z$ with $\mathcal{D}_z^{RL}$.
23:     **end if**
24: **end for**

---

to verify this claim, we first evaluate the diversity of coordination policies learned by SPD and URL baselines in Multi-agent Particle Environment[2] [22, 45]. We customize the scenario *SimpleTag*, which is a predator-prey environment, for URL experiments by discarding the extrinsic reward from the environment. Instead, we are concerned about the diversity of agents' formations driven by learned coordination policies. The detailed description of the scenario please refer to Appendix B.1.

**Baselines and evaluation**     We compare our method SPD with several conventional single-agent URL approaches. Here is the details about the implementation of URL algorithms, which we carried out to learn policies without external reward:

- SPD: we follow the narration of the implementation details in Sec. 4.4. The pseudo-reward is derived via Eq. (1)~(7).

- Diversity Is All You Need (DIAYN) [10]: The joint policy $\boldsymbol{\pi}(\boldsymbol{a}|\boldsymbol{\tau}_z, z)$ is conditioned on a discrete latent variable $z$ sampled from a pre-defined categorical distribution $p(z)$: $z \sim p(z)$. The pseudo-reward with formulation as $r_z := \log q_\phi(z|\{o^i\}_{i=1}^n) - \log p(z)$ is used for inspiring agents to visit states that can be distinguished easily, where $q_\phi(z|\{o^i\}_{i=1}^n)$ is a learned discriminator. In our experiment, we replace the actor network $\boldsymbol{\pi}(\boldsymbol{a}|\boldsymbol{\tau}_z, z)$ shared by all coordination policies with $Z$ individual networks $\{\boldsymbol{\pi}_z\}_{z=1}^Z$, considering the large observation space and action space.

- Wasserstein Unsupervised Reinforcement Learning (WURL) [15]: The pseudo-reward is designed to enhance the distance between different policies, and it is obtained by estimating the Wasserstein distance of the induced state distributions: $r_z = W(\boldsymbol{\tau}_z, \bar{\boldsymbol{\tau}}_k)$, where $\boldsymbol{\tau}_z$ is the trajectory driven by current joint policy $\boldsymbol{\pi}_z(\boldsymbol{a}|\boldsymbol{\tau}_z)$ and $\bar{\boldsymbol{\tau}}_k$ is a sampled batch from another joint policy $\boldsymbol{\pi}_k$.

---
[2]https://github.com/Farama-Foundation/PettingZoo/tree/master/pettingzoo/mpe

The URL algorithms are implemented with QMIX to learn diverse joint policies $\{\boldsymbol{\pi}_z\}_{z=1}^Z$, and we set $Z = 10$ in practice. The hyper-parameters are kept to be the same, and please refer to Appendix B.2 for details. For evaluation, each algorithm is carried out with 5 different random seeds and trained with $4 \times 10^6$ steps interacting with the environment. All the intrinsic pseudo-rewards are derived based on the positions of agents, which is a part of the original observation space.

Since DIAYN and WURL are designed for the single agent case, we deploy them by regarding all agents as a single agent. Concretely, we use observations from all agents to create the features that DIAYN and WURL need.

| Method | DSR(%) | Normalized $\bar{d}_{sp}$ |
|---|---|---|
| DIAYN | $59.9 \pm 0.9$ | $(4.0 \pm 2.7) \times 10^{-2}$ |
| WURL | $58.7 \pm 1.5$ | $(3.9 \pm 3.6) \times 10^{-3}$ |
| SPD (Ours) | $58.6 \pm 0.7$ | $1.0 \pm 0.1$ |

Table 1: The discriminator success rate (DSR) and normalized proximal discrepancy of synergy patterns $\bar{d}_{sp}$ for URL methods DIAYN, WURL and SPD (Ours).

We evaluate these algorithms following the same protocol: (i) collect sample transitions except extrinsic reward using each joint policy $\boldsymbol{\pi}_z(\boldsymbol{a}|\boldsymbol{\tau}_z)$, (ii) calculate the pseudo-rewards $\{r_z\}$ for each time step and then assign them to the trajectory $\boldsymbol{\tau}_z$, (iii) employ learned joint policies once the training procedure is completed to collect $\{\boldsymbol{\tau}_z^{\text{final}}\}_{z=1}^Z$ for evaluation. We use two metrics to evaluate how diverse are the learned joint policies from different aspects: *the diversity of states* and *the discrepancy among relative relationships of agents* induced by $\{\boldsymbol{\pi}_z\}_{z=1}^Z$. As for the former one, we adopt the structure of the discriminator $q_\phi(z|\{o^i\}_{i=1}^n)$ from DIAYN, which is a learned neural network to classify each policy $\boldsymbol{\pi}_z$ from state samples in $\{\boldsymbol{\tau}_z^{\text{final}}\}_{z=1}^Z$, and use the success rate of the discriminator to measure how diverse are the states, namely Discriminator Success Rate (DSR). As discussed in Sec. 4.2, the discrepancy of synergy patterns $d_{sp}$ according to Eq. (2) is an effective measure for the latter one. Table 1 shows that the performance of all algorithms on DSR is close, whereby SPD significantly outperforms the others on the discrepancy of synergy patterns. Such results demonstrate that SPD can inspire agents to visit comparably identifiable states as the conventional URL approaches, while further motivating agents to explore diverse coordination policies in the meanwhile.

**Ablation study** We examine the following components of our method in detail to show their effects: (i) Reward scale $\beta_r$: we multiply the pseudo-reward in Eq. (5) with a factor vary in $\{0.1, 1.0, 10.0\}$ to change the scale of the final returns received by agents, and 10.0 is the default value in the aforementioned experiments. (ii) Sparse return $\tilde{R}_z$: To evaluate the effect of the decomposition of the pseudo-return in Eq. (4), we directly use it as the pseudo-reward of the last step in the episode. (iii) On-policy $\rho(t) = t$: As discussed in Sec. 4.2 that the SPG from $\bar{\lambda}^{\boldsymbol{\pi}_2}$ corresponding to $G_t^{\boldsymbol{\pi}_1}$ may not be $G_t^{\boldsymbol{\pi}_2}$. To verify this claim, we set $\rho(t) = t$ directly and the pseudo-reward becomes to $\tilde{r}_t^{\text{(no match)}}(\boldsymbol{\tau}_z, \{\boldsymbol{\tau}_k\}) = d_{gw}(G_t^{\boldsymbol{\pi}_z}, G_t^{\boldsymbol{\pi}_{k*}})$, where $G_t^{\boldsymbol{\pi}_z} \in \bar{\lambda}^{\boldsymbol{\pi}_z}, G_t^{\boldsymbol{\pi}_{k*}} \in \bar{\lambda}^{\boldsymbol{\pi}_{k*}}$. (iv) Sub-optimal $\bar{\rho}$: The matching function $\rho^*$ is obtained by Kuhn–Munkres algorithm [28]. Therefore, the precision is influenced by the number of iterations, which defaults to $\mathcal{N}_{\text{KM}}^{\text{iter}} = 100$ in the experiments. A lower number of iterations $\mathcal{N}_{\text{KM}}^{\text{iter}} = 25$ is used to show the influence from the precision of the matching function.

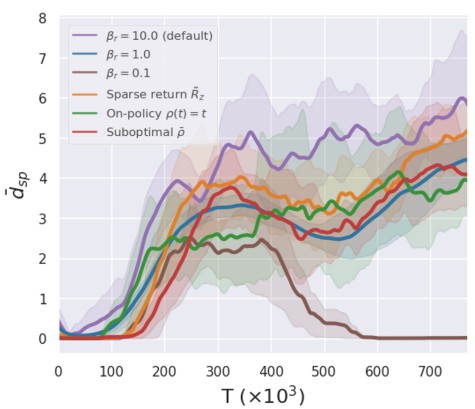

Figure 2: Ablation study. We change the scaling factor $\beta_r$ multiplying the pseudo-reward from [0.1, 1.0, 10.0]. "Sparse return $\tilde{R}_z$" means using Eq. (4) as final the reward directly. "On-policy $\rho(t) = t$" means we apply no optimal transport and set $\rho(t) = t$ in Eq. (3) to suit the on-policy manner computing pseudo-reward on-the-fly. And we reduce the precision of the solution for Eq. (6) in "Sub-optimal $\bar{\rho}$".

We use $\bar{d}_{sp}$ as the evaluating indicator and the results is shown in Fig. 2. The curves start from the time step that the exploration rate epsilon has stopped annealing. We can summarize that the reward scale plays an important role in learning good synergy patterns. Canceling the optimal transport for

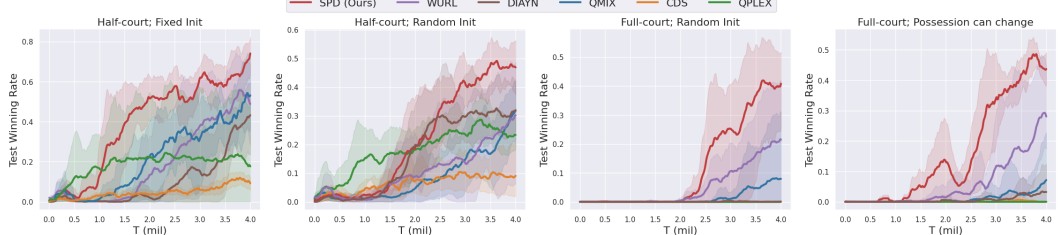

Figure 3: Compare the performances of using one model trained by URL algorithms as the parameter initialization for four different tasks with learning by traditional MARL algorithms directly.

matching or decreasing its accuracy both harm the performance. And the "Sparse return $\tilde{R}_z$" curve shows that amortizing the episode return to each time step simplifies the training.

## 5.2 Generalizability of learned synergy patterns

In this part we demonstrate the generalization ability of the models trained by SPD. Specifically, we first train SPD on the complicated MARL environment: Google Research Football [18] without environment reward. Then we use the trained model as the initialization for different tasks with specific rewards, and use common MARL algorithms to train it.

**Task description** We choose the map "academy_3_vs_1_with_keeper" with the setting from CDS [5], where three players need to cooperate to shoot and get a score under the defense of an opposing player and a goalkeeper. To simplify the training, the original environment ends an episode if the ball or the player return to the left half-court. Players always start at the same place. Losing the control of the ball also cause the failure. The team is rewarded 100 at the end of an episode when there is a goal, otherwise $-1$. On the basis of this scenario which we call "Half-court; Fixed Init", we design a series of tasks with more difficulty. (i) We initialize the agents randomly in a circle with 5% court width's radius, which increases the difficulty by introducing more randomness. (ii) We remove the restrictions on the left-half court, which greatly increases the exploration space. (iii) The possession change will not end the episode, which may increase the percent of the bad samples in the replay buffer, since at the beginning of training the ball is easily robbed by the preset AI opponent.

**Utilizing SPD model** We first train the models by SPD with $Z = 20$ on the "Half-court; Possession can change" scenario without the environment reward for $10^6$ time steps, as well as WURL [15] and DIAYN [10]. Then we use the same trained parameters as the initialization for QMIX [34] on four different downstream scenarios. In contrast to the manual selection of the synergy patterns, we test them on the downstream task before MARL training, choosing the parameters of the model with the highest average return as the model initialization. We show the results in Fig. 3, together with the SOTA on GRF including QMIX [34], QPLEX [46] and CDS [5]. The results show that using the model learned by conventional single-agent URL methods (WURL and DIAYN) as the initialization mostly performs similarly to the baseline QMIX, while WURL slightly outperforms QMIX on the "Full-court" maps. In contrast, SPD learns faster and finally gets higher winning rate on all scenarios, demonstrating the effectiveness of $d_{sp}$ as an evaluation metric for the difference of agents' relationship and showing that SPD embeds the model with stronger generalization ability. Actually, one difference between SPD and these URL methods is that SPD encourages the diversity of the relationship among agents while these diversity methods encourages the diversity of visited states. Therefore, the models trained by WURL may reach diverse states on the 'Full-court' maps which need more exploration and we believe this accounts for the slightly better performance.

**Visualization of the learned synergy patterns** To comprehensively evaluate SPD, we visualize[3] the trajectories of all the skills and show three of them in Fig. 4. The trajectories of other skills can be found in Appendix C. We find that agents can perform some meaningful behavior from a human perspective, even without the supervision of the environment reward. Agents can dribble for a long

---

[3]The videos for all synergy patterns can be found at: `https://sites.google.com/view/spd-umarl`

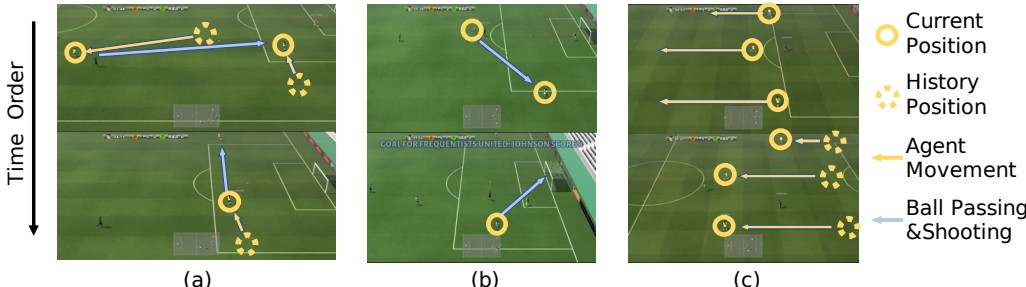

Figure 4: Visualization of the learned synergy patterns without environment reward. (a) Dribbling and long passing. (b) Passing and shooting. (c) Running back to play defense.

distance, pass between teammates, shoot, and run back together in different skills. To our concern that the different performance of different joint policies learned by SPD are only due to the randomness of the GRF environment itself but not the inter-policy diversity we want, we deploy each learned joint policy for 10 episodes and visualize them. The results show that each joint policy exhibits the similar coordination behavior across episodes. The visualization shows that making the team coordinate more diverse drives agents to learn meaningful behaviour, which is favorable for downstream tasks.

## 6 Discussion

### 6.1 Limitations

There are still some limitations or difficulties with further applications. Though the discriminator success rate (DSR) used in Sec. 5.1 is not a good measure for relationships of agents, it does evaluate the diversity of visited states in some ways. The results on DSR show that all of the URL approaches, including our method SPD, are far away from satisfying, implying that there is still space for improvement. Furthermore, a synergy pattern function $\zeta$ that embeds more information than the one used in this paper is tend to require higher computational cost. Moreover, incorporating diversity-based exploration technique with SPD may improve the intra-agent diversity as well as inter-policy diversity. These limitations may suggest further research directions.

### 6.2 Negative societal impact

Due to the domain gap between virtual environments and real-world scenarios, applying our method directly to some real-world applications, such as traffic management, is risky and may result in damage to the system. To avoid harmful operations, human supervision is required to restrict the learned models.

## 7 Conclusion

In this work, we propose a general unsupervised framework *Synergy Pattern Diversifying Oriented Unsupervised Multi-agent Reinforcement Learning* (SPD) to learn useful synergy patterns for agents. Synergy pattern graph (SPG) is devised to depict the relationship among all agents. Furthermore, we approximate the discrepancy of synergy patterns by applying optimal transport on the batches sampled from the distributions of SPG. In our experiments, we show that SPD surpasses conventional URL approaches on capturing the relative relations of agents and learns useful synergy patterns with no task-related reward. We also empirically demonstrate that the learned policies can accelerate the downstream training procedure and outperform state-of-the-art algorithms in GRF.

## 8 Acknowledgement

This work was supported by the National Key R&D Program of China under Grant 2018AAA0102801, National Natural Science Foundation of China under Grant 61620106005.

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
