# A Properties of the discrepancy of synergy patterns as a legitimate pseudometric

**Proposition A.1.** *By the definition 4.1, $d_{sp}$ is a legitimate pseudometric on $\Lambda$ which can be proved by examining following properties:*

*i)* $d_{sp}(p_1, p_2) \geq 0$,

> *Proof.* As Chowdhury & Mémoli [1] show, the Gromov-Wasserstein discrepancy is a pseudo-metric on $\mathcal{G}$, where $\mathcal{G}$ denotes the collection of measure graphs, leading to $d_{gw}(G^{\pi_1}, G^{\pi_2}) \geq 0$. Following the definition in Eq. (2), proving $d_{sp}(p_1, p_2) \geq 0$ is trivial since all elements are non-negative. □

*ii)* $d_{sp}(p, p) = 0$,

> *Proof.* Similarly, based on $d_{gw}(G^{\pi}, G^{\pi}) = 0$, we can choose $\varrho$ carefully as $\varrho_0$ with $\mathbb{P}(G^{\pi_1}, G^{\pi_2} = G) = \delta(G)$ and $\mathbb{P}(G^{\pi_1} = G, G^{\pi_2}) = \delta(G)$ everywhere, where $\delta$ is the Dirac distribution. Hence we have:
> $$d_{sp}(p, p) = \left[ \int d_{gw}(G^{\pi_1}, G^{\pi_2}) \, \mathrm{d}\varrho \right]_{\varrho = \varrho_0} = \int 0 \cdot \mathrm{d}\varrho = 0$$
> □

*iii)* $d_{sp}(p_1, p_2) = d_{sp}(p_2, p_1)$,

> *Proof.* To avoid unnecessary confusion, we notate the joint distribution of the L.H.S as $\varrho_l$ and that of the R.H.S. as $\varrho_r$, respectively. Denote $\varrho_l = \varrho_{12}$ as one of the joint distributions when the infimum of the L.H.S. is reached:
> $$d_{sp}(p_1, p_2) = \left[ \int d_{gw}(G^{\pi_1}, G^{\pi_2}) \, \mathrm{d}\varrho_l \right]_{\varrho_l = \varrho_{12}}$$
> We show that the infimum of the R.H.S. is reached when $\varrho_r = \varrho_{21}$ by contradictions, where $\varrho_{21}$ can be obtained by exchanging the probabilities of $G^{\pi_1}$ and $G^{\pi_2}$ in $\varrho_{12}$. Suppose there exists $\varrho_r = \varrho'_{21}$ such that:
> $$\left[ \int d_{gw}(G^{\pi_2}, G^{\pi_1}) \, \mathrm{d}\varrho_r \right]_{\varrho_r = \varrho'_{21}} < \left[ \int d_{gw}(G^{\pi_2}, G^{\pi_1}) \, \mathrm{d}\varrho_r \right]_{\varrho_r = \varrho_{21}}$$
> Then we can update the joint distribution for the L.H.S. with $\varrho_l = \varrho'_{12}$ by exchanging the probabilities of $G^{\pi_1}$ and $G^{\pi_2}$ in $\varrho'_{21}$ similarly. Based on the property of the Gromov-Wasserstein discrepancy: $d_{gw}(G^{\pi_1}, G^{\pi_2}) = d_{gw}(G^{\pi_2}, G^{\pi_1})$, we have:
> $$\left[ \int d_{gw}(G^{\pi_1}, G^{\pi_2}) \, \mathrm{d}\varrho_l \right]_{\varrho_l = \varrho'_{12}} = \left[ \int d_{gw}(G^{\pi_2}, G^{\pi_1}) \, \mathrm{d}\varrho_r \right]_{\varrho_r = \varrho'_{21}}$$
> $$< \left[ \int d_{gw}(G^{\pi_2}, G^{\pi_1}) \, \mathrm{d}\varrho_r \right]_{\varrho_r = \varrho_{21}}$$
> $$= \left[ \int d_{gw}(G^{\pi_1}, G^{\pi_2}) \, \mathrm{d}\varrho_l \right]_{\varrho_l = \varrho_{12}}$$
> This contradicts the claim that $\varrho_l$ is one of the joint distributions when the infimum of the L.H.S. is reached. Hence, we can derive:
> $$d_{sp}(p_2, p_1) = \inf_{\varrho_r \in \Gamma[p_2, p_1]} \int d_{gw}(G^{\pi_2}, G^{\pi_1}) \, \mathrm{d}\varrho_r$$
> $$= \left[ \int d_{gw}(G^{\pi_2}, G^{\pi_1}) \, \mathrm{d}\varrho_r \right]_{\varrho_r = \varrho_{21}}$$
> $$= \left[ \int d_{gw}(G^{\pi_1}, G^{\pi_2}) \, \mathrm{d}\varrho_l \right]_{\varrho_l = \varrho_{12}}$$
> $$= d_{sp}(p_1, p_2)$$

| Name | Description | Value |
|---|---|---|
| $\gamma$ | Discounted factor | 0.99 |
| $\varepsilon$ anneal time | Time-steps for $\varepsilon$ to anneal from $\varepsilon_\mathrm{s}$ to $\varepsilon_\mathrm{f}$, where $\varepsilon$ is the probability for agents choosing random actions. | 100000 |
| $\varepsilon_\mathrm{s}$ | Initial $\varepsilon$ at start | 1 |
| $\varepsilon_\mathrm{f}$ | Final $\varepsilon$ | 0.05 |
| $\mathcal{N}_\mathrm{env}$ | The number of parallel environments | 1 |
| $|\mathcal{D}^{RL}|$ | The size of the replay buffer for MARL learning | 5000 |
| $\mathcal{N}_\mathrm{rnn}$ | Dimension of RNN cells | 256 |
| $lr$ | Learning rate | 0.001 |
| $\mathcal{N}_\mathrm{batch}$ | Batch size | 128 |
| $t_\mathrm{target}$ | Time interval for updating the target network | 200 |
| $G_\mathrm{max}$ | Clipping value for all gradients | 10 |
| $|\mathcal{D}|$ | The size of the graph buffer for SPD | 10000 |
| $t_\mathrm{start}$ | The start steps for employing SPD to obtain pseudo-reward | 5000 |
| $\alpha$ | The factor of the regularized term in Eq. (6) | 0 |
| $B_1, B_2$ | The size of the SPG batches | 50 |
| $\mathcal{N}_\mathrm{Sinkhorn}^\mathrm{iter}$ | The number of the iterations for Sinkhorn-Knopp algorithm | 50 |
| $\mathcal{N}_\mathrm{KM}^\mathrm{iter}$ | The number of the iterations for Kuhn–Munkres algorithm | 100 |

Table 1: Hyper-parameters.

which finishes the proof. $\qquad\square$

*iv)* $d_{sp}(p_1, p_\chi) + d_{sp}(p_2, p_\chi) \geq d_{sp}(p_1, p_2)$.

*Proof.* We prove the triangle inequality by contradictions similar to iii). Suppose that $\exists p_\chi$, s.t. $d_{sp}(p_1, p_\chi) + d_{sp}(p_2, p_\chi) < d_{sp}(p_1, p_2)$. Denote the joint distributions for the infimum of the discrepancy of synergy patterns as $\varrho_{1\chi}, \varrho_{2\chi}$ and $\varrho_{12}$ respectively. Let $\varrho_{\chi2}$ be the joint distribution by exchanging the probability of $G^{\boldsymbol{\pi}_2}$ and $G^{\boldsymbol{\pi}_\chi}$. Then we can use $\varrho_{1\chi}$ and $\varrho_{\chi2}$ to find the probabilities of $G^{\boldsymbol{\pi}_1}$ and $G^{\boldsymbol{\pi}_2}$ corresponding to the same $G^{\boldsymbol{\pi}_\chi}$, obtaining the new joint distribution $\varrho_{12}'$. By the triangle inequality of the Gromov-Wasserstein discrepancy, we can write:

$$
\begin{aligned}
\left[\int d_{gw}(G^{\boldsymbol{\pi}_1}, G^{\boldsymbol{\pi}_2})\, \mathrm{d}\varrho\right]_{\varrho=\varrho_{12}'} &\leq \left[\int d_{gw}(G^{\boldsymbol{\pi}_1}, G^{\boldsymbol{\pi}_\chi})\, \mathrm{d}\varrho\right]_{\varrho=\varrho_{1\chi}} + \left[\int d_{gw}(G^{\boldsymbol{\pi}_\chi}, G^{\boldsymbol{\pi}_2})\, \mathrm{d}\varrho\right]_{\varrho=\varrho_{\chi2}} \\
&= \left[\int d_{gw}(G^{\boldsymbol{\pi}_1}, G^{\boldsymbol{\pi}_\chi})\, \mathrm{d}\varrho\right]_{\varrho=\varrho_{1\chi}} + \left[\int d_{gw}(G^{\boldsymbol{\pi}_2}, G^{\boldsymbol{\pi}_\chi})\, \mathrm{d}\varrho\right]_{\varrho=\varrho_{2\chi}} \\
&= d_{sp}(p_1, p_\chi) + d_{sp}(p_2, p_\chi) \\
&< d_{sp}(p_1, p_2)
\end{aligned}
$$

which contradicts the claim that $d_{sp}(p_1, p_2)$ is the infimum. Hence we have $\forall p_\chi, d_{sp}(p_1, p_\chi) + d_{sp}(p_2, p_\chi) \geq d_{sp}(p_1, p_2)$. $\qquad\square$

# B  Details for experiments and reproducibility

## B.1  Description for the MPE environment in Sec. 5.1

Multi-agent Particle Environment (MPE) [3, 6] consists of $n$ agents and $l$ landmarks in a 2D world. Each agent has to resolve to select the action from its discrete action space to move around. In practice, the experiments in Sec. 5.1 are carried out in the customed scenario *SimpleTag*, which is a predator-prey environment with $n = 4, l = 1$. In the original scenario, there are 1 good agent that can move faster to keep itself away from the others and 3 adversaries trying to hit the good agent. Besides, there is a obstacle blocking the way denoted by 1 landmark. To customize the environment for URL experiments, we discard the extrinsic reward from the environment. Instead, we are concerned about the diversity of agents' formations driven by learned coordination policies.

## B.2  Details about hardware and reproducibility

**The hardware**  An AMD Ryzen 3975WX CPU with 32-Cores and three RTX-3090-11G GPUs are employed to run all the experiments with five random seeds. As for MPE, it takes around 10 hours for

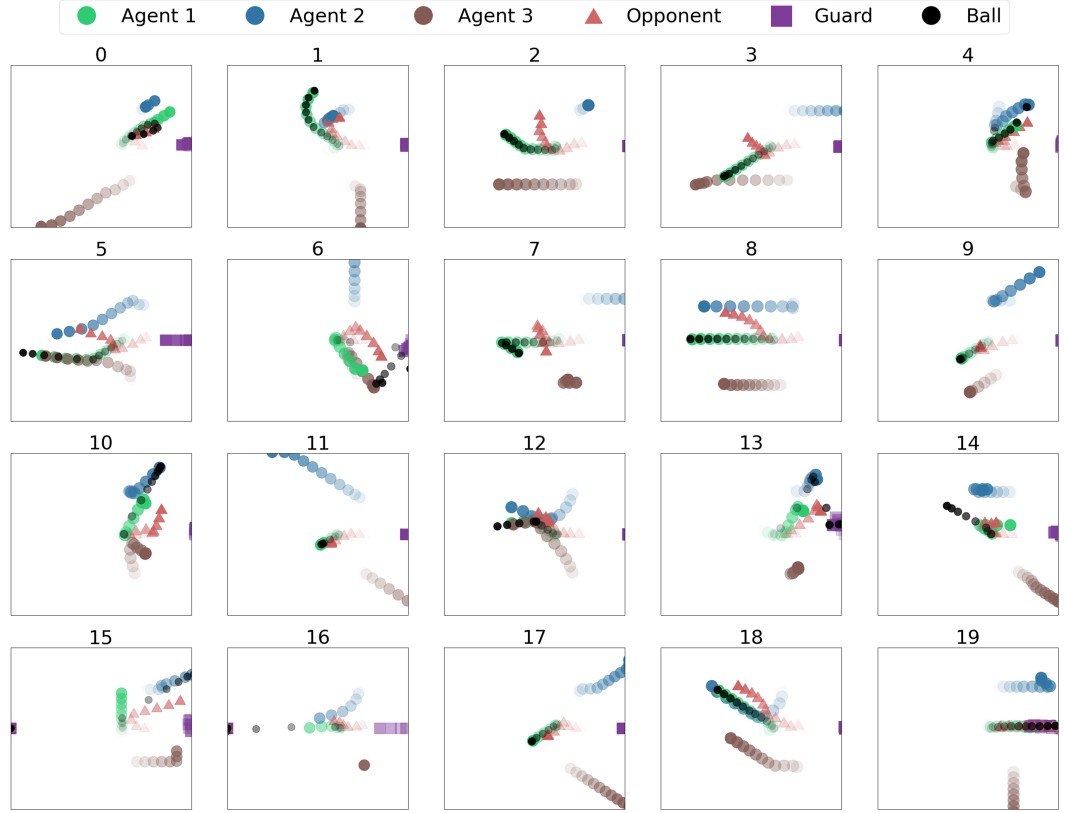

Figure 1: Position trajectories of $Z = 20$ synergy patterns, and the number of each sub-figure is the id $z$ of the synergy pattern. The transparency of the circle denotes the time in the episode and the more transparent circle is at the smaller time step.

$10^6$ time steps to train 10 joint policies with SPD. Besides, SPD takes around 36 hours for $4 \times 10^6$ time steps in GRF to train 20 joint policies.

**Code and reproducibility**   Our entire code [1] is opened for reproducing all the experiments. To reproduce the results, please refer to the instruction file "README.md".

**Training details**   We adopt QMIX [4] as our MARL algorithm to learn synergy patterns. Recurrent Neural Network (RNN) is used in the policy to alleviate the partial observability. The mixing network has one hyper-layer as described in QMIX with 64 units. The optimizer to optimize the neural networks is "Adam". Each URL algorithm is deployed to learn different joint policies ($Z = 10$ for MPE and $Z = 20$ for GRF) and mixing networks every time. We summarize most of the hyper-parameters for the two experiments in Sec. 5 in Table. 1.

## C   Visualization for learned synergy patterns in GRF

To comprehensively illustrate the diversity of synergy patterns learned by SPD, we directly visualize the $Z = 20$ learned joint policies in GRF. During this procedure, The agents $1, 2, 3$ are controlled by the learned policies and the opponents are built-in AI. Fig. 1 shows the position trajectories of the agents and the opponents by different learned joint policies. The results demonstrate that the agents can learn useful synergy patterns, such as dribbling ($z = 1, 2, 3, 7, \dots$), collaborating to maintain the formations ($z = 4, 8, 18$), and passing-and-shooting ($z = 6$), with only pseudo-reward from SPD.

---

[1] We custom the code from `https://github.com/hijkzzz/pymarl2` [2] to carry out the experiments in this paper.

| Name | Description |
|---|---|
| Synergy Pattern | The perennial coordinated behaviour of agents. |
| Synergy Pattern Graph (SPG) $G^{sp}(\mathcal{V}, \boldsymbol{W}, \boldsymbol{\mu}, \zeta)$, a.k.a. SPG element | A graph, where $v_i \in \mathcal{V}$ is the vertex for agent $i \in \mathcal{I}$ and the weight $\omega_{ij} \in \boldsymbol{W}$ of edge $\{i, j\}$ depicts agents' relative relations. |
| Synergy Pattern Function $\zeta$ | A general function which could depict agents' relative relations. |
| SPG batch $\bar{\lambda}$ | A batch of SPG sampled from the distribution of SPG. |
| Discrepancy of Synergy Patterns $d_{sp}$ | The discrepancy between two distributions of SPG. |
| Proximal Discrepancy of Synergy Patterns $\bar{d}_{sp}$ | The sum of the $d_{gw}$ between two SPG batches. |

Table 2: Concepts in SPD.

# D Discrimination of proposed concepts

We summarize the concepts mentioned in this work in Table 2 for a clearer understanding.

# E Evaluation on SMAC

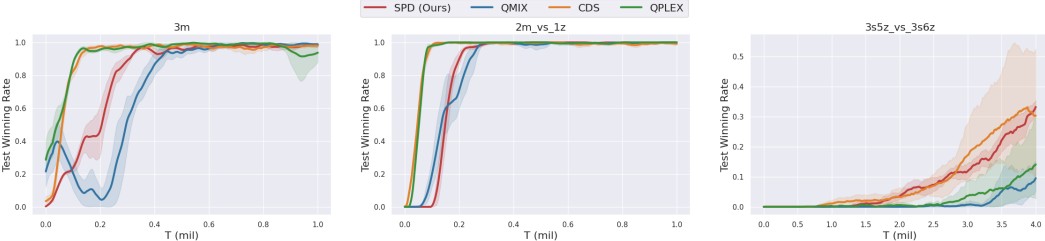

Figure 2: Comparison of our approach SPD against baseline algorithms on three SMAC maps.

We also assess SPD on SMAC [5] to demonstrate the efficacy of the learnt joint policies and offer a more thorough perspective of the effect of the URL training. As for our method SPD, we follow the same URL training procedure in Sec. 5.2 and use QMIX as our training method as well. The $Z = 20$ pre-trained policies are learned by QMIX on the same maps as the downstream tasks without external reward but only pseudo-reward provided by SPD for $10^6$ time steps, and are tested before MARL training on the downstream tasks to choose parameters as the initialization. During the MARL training, each algorithm is carried out with 5 random seeds.

We evaluate each algorithm on three maps: 3m, 2m_vs_1z and 3s5z_vs_3s6z (super-hard). The results are shown in Fig. 2. On the map 2m_vs_1z with normal difficulty, SPD performs similarly to the baseline QMIX, while it learns more efficiently compared to QMIX on map 3m. We believe the fact that these two maps can be explored by $\epsilon$-greedy strategy sufficiently and do not need well-performed team behaviour may account for this. Since we use QMIX as our training algorithm, the performance is limited by it and is surpassed by CDS and QPLEX. However, SPD significantly outperforms the baseline QMIX and reach similar performance as CDS on the super-hard map 3s5z_vs_3s6z, which demonstrates that SPD do learn the relationship of agents and encourage team behaviour. Such results suggest that the role of SPD is more pronounced in tasks where there is a greater need for cooperation. In addition, combining exploration-based algorithms, such as CDS, with SPD may achieve better performance and foreshadow future research directions.