# OpenReview forum: "SPD: Synergy Pattern Diversifying Oriented Unsupervised Multi-agent Reinforcement Learning"
_NeurIPS.cc/2022/Conference — NeurIPS 2022 Accept_

### Official Review · Reviewer_8Kca · 2022-07-09

**Rating:** 6
**Confidence:** 4
**Soundness:** 3 good
**Presentation:** 3 good
**Contribution:** 3 good

**Summary:**

The paper presents a method for unsupervised multi-agent reinforcement learning which uses diversity bonuses as an intrinsic reward.  The SPD method rewards diversity between _synergy pattern graphs_, which measure the similarity of the full set of agents' behavior.  The paper argues that diversity as measured between these graphs can capture the dependencies between agents better than promoting diversity in the state space alone.  The paper demonstrates improved performance in the Google Research Football (GRF) benchmark.

# Update after author response

Thank you for the well-considered responses and updated experiments.  The updated results provide a stronger support for the SPD method, and I have raised my score to reflect them.

**Questions:**

Related to my concern above, what do you expect the relationship between $d_{sp}$ and downstream task performance to be?  Did I misunderstand why it should be a good metric on its own?  My rating depends mostly on this point, because the approach seems promising and the results are good on one benchmark, but it could be possible to achieve these results with more straightforward diversity methods.  Arguments in the paper suggest reasons why we should expect DIAYN and WURL to perform worse, but there is no experiment to back this up.  Since this is a major claim of the paper, I'd like to see it supported.

**Limitations:**

The authors address the limitations and potential societal impact well in the appendix.

**Strengths And Weaknesses:**

### Originality
 - To the best of my knowledge, the SPD method is a novel approach that combines diversity-based unsupervised RL approaches with a graph-based representation of agent interactions.

### Quality
 - The comparisons between SPD, DIAYN and WURL in section 5.1 use $d_{sp}$ as an evaluation metric, which seems incomplete.  Without establishing that $d_{sp}$ correlates with, say, better down-stream task generalization performance, it is difficult to be confident in conclusions drawn using it as a metric.  I would have liked to see these baselines (DIAYN and WURL) used in the experiments in section 5.2 as well.  Can we attribute the improved performance of SPD demonstrated in figure 3 to diversity bonuses in general, or specifically to the diversification of synergy patterns?

### Clarity
 - Although the SPD method is complex, the authors do a good job describing the approach, including describing the necessary background for understanding SPD.
 - The discussion of the ablation experiments in section 5.1 and figure 2 is very light on details.  I am not sure what the reward scale is, what the precision on the solution means, and why these are relevant ablations to consider.
 - A diagram illustrating $G^{sp}$ and $d_{sp}$ could aid understanding, even though the descriptions are clear.

### Significance
The significance of the approach is hard to judge given the current experiments.  The approach seems to make advances on the state of the art in CTDE MARL approaches on GRF, but the concerns raised in the Quality section of this review above make it hard to know what to attribute the performance to.

---

> ### Author Response · Authors · 2022-08-02
> **Response to Reviewer 8Kca**
>
> We appreciate your review and constructive suggestions.
> We hope the following answers your questions and addresses your concerns.
>
> > The comparisons between SPD, DIAYN and WURL in section 5.1 use $d_{sp}$ as an evaluation metric, ... the diversification of synergy patterns?
>
> We are inspired by your important advice.
> We have to admit that when designing the experiments, we took for granted that these methods, which were not designed for multi-agents, would have difficulty capturing the relationship of the agents, and ignored this part of the experimental comparison.
> The further experiments of the DIAYN and WURL on GRF (alse based on QMIX with the same setting as SPD) are carried out, and we have updated the results in Fig. 3 in the *Rebuttal Revision*.
>
> The results show that using the model learned by conventional URL methods (WURL and DIAYN) as the initialization mostly performs similarly to the baseline QMIX, while WURL slightly outperforms QMIX on the 'Full-court' maps.
> In contrast, SPD learns faster and finally gets higher winning rate on all scenarios which demonstrate the effectiveness of $d_{sp}$ as an evaluation metric.
> Actually, one difference between SPD and these URL methods is that SPD encourages the diversity of the relationship among agents while these diversity methods encourages the diversity of visited states.
> Therefore, the models trained by WURL may reach diverse states on the 'Full-court' map which need more exploration and we believe this accounts for the slightly better performance.
>
> > The discussion of the ablation experiments in section 5.1 and figure 2 is very light on details. I am not sure what the reward scale is, what the precision on the solution means, and why these are relevant ablations to consider.
>
> Due to the limits of the pages by NeurIPS, there is not enough space to describe the ablation experiments in detail in the main body.
> Therefore we delineate the ablation experiments in Appendix B.1.
> We also update the descriptions to make it more clear in the *Rebuttal Revision*.
>
> 'Reward scale $\beta_{r}$' is a constant factor multiplying the pseudo-reward in Eq. (5) to change the order of magnitude of it, since the neural network might be insensitive to the original magnitude of pseudo-reward (like $1\times 10^{-1}$).
>
> 'Sparse return $\tilde{R}_z$' removes the decomposition of the pseudo-return, which means using $\tilde{R}_z$ in Eq. (4) directly as the final reward.
>
> The other two setting is about the process of solving for the optimal matching function $\rho^*$ in Eq. (3).
> 'On-policy $\rho(t)=t$' removes this process entirely and simply sets $\rho(t)=t$, while 'Suboptimal $\bar{\rho}$' uses a lower number of iterations in this process and obtains a suboptimal matching funcion $\bar{\rho}$.
>
> > A diagram illustrating $G^{sp}$ and $d_{sp}$ could aid understanding, even though the descriptions are clear.
>
> ### **Update**
> We are delighted to inform that we illustrate $G^{sp}$ and $d_{sp}$ in Fig. 1 in the newest revision, and we hope this will adress your conern.
>
> Thanks to your valuable comments again.
>
> > Q1. Related to my concern above, what do you expect ... it supported.
>
> Briefly, we expect that learned joint policies with high $d_{sp}$ will learn faster in downstream tasks.
>
> To this point, an important hypothesis made by DIAYN is that the higher the coverage of learned skills over the set of possible behaviour, the greater the probability of gaining effective skills (Line 34-36).
> Basically, we devise the discrepancy of synergy patterns $d_{sp}$ to evaluate the difference between two joint policies in multi-agent setting.
> And increasing the $d_{sp}$ aims to improve the coverage of learned joint policies over the set of possible coordination behaviour.
> When some policies keep high $d_{sp}$ to other useless policies, these policies may learn useful coordination behaviour.
> As the results of the visualization on GRF show (Fig. 4, and videos can be found at our [site](https://sites.google.com/view/spd-umarl)), some synergy patterns seems to be meaningless (such as kick the ball directly out of bounds), while some synergy patterns are surprising (such as pass-and-shoot).
> The learned policies with these useful synergy patterns are intuitively be good initializations for the downstream tasks, leading to efficient downstream learning.
> Also, our experimental results demonstrate that SPD does learn faster than the original baseline QMIX.
>
> As for the assumption that "it could be possible to achieve these results with more straightforward diversity methods", we have shown the performance of URL approaches on GRF and discussed it in the former part.

---

### Official Review · Reviewer_meoo · 2022-07-09

**Rating:** 6
**Confidence:** 3
**Soundness:** 3 good
**Presentation:** 3 good
**Contribution:** 3 good

**Summary:**

The authors proposed a MARL algorithm to learn generic coordination policies for agents with no extrinsic reward called Synergy Pattern Diversifying Oriented Unsupervised MARL (SPD). They utilized a graph representing the relationships of agents at each time step called Synergy Pattern Graph (SPG), and an episode-wise divergence measurement to approximate the discrepancy of synergy patterns. Results showed the capacity of SPD to acquire meaningful coordination policies, such as maintaining specific formations in Multi-Agent Particle Environment and pass-and-shoot in Google Research Football. Furthermore, they demonstrated that the same instructive pretrained policy’s parameters can serve as a good initialization for a series of downstream tasks’ policies, achieving higher data efficiency and outperforming state-of-the-art approaches in Google Research Football.

**Questions:**

* L 84: no extra cost?
* L 244: utilize/an assignment?
* Algorithm 1 L15: Is eq. (2) correctly eq. (3)?
* The performances of CDS (Li et al. 2021, NeurIPS) in Figure 3 may be worse than expected (in the paper, CDS outperformed QMIX and QPLEX). I previously ran their code and it worked well. I know the experimental condition was different, but I want to know the reason (were there no code and hyperparameters for CDS?).
* There was no video at https://sites.google.com/view/spd-umarl.
* L328-329: “To avoid the effect of randomness, we repeat the visualization for 10 times for each skill.” It was not clear to me. What did the authors do?
* Appendix L83: The results show that none of the URL approaches, including our method SPD, achieve more than 75% on DSR. The values in Fig. 2 a seem to be 60%. Why did the authors mention 75%?


**Limitations:**

The limitations and potential negative societal impact were described in Appendices D and E, respectively.


**Strengths And Weaknesses:**

The strength of this paper is as follows:
* The proposed unsupervised MARL algorithm seems to be original, and they used the synergy graph to train team skills without rewards from the environment, which brings the agents good generalization ability in downstream tasks, and takes no extra cost in execution.
* The experimental results demonstrate that SPD achieves better performance in MARL compared to conventional unsupervised RL approaches and shows great potential to learn synergy patterns with generalizability for downstream tasks.

The weakness of this paper is as follows:
* The presentation was sometimes unclear (please see below)
* The experiment descriptions were sometimes unclear (please see below)

---

> ### Author Response · Authors · 2022-08-02
> **Response to Reviewer meoo**
>
> Thanks for your feedback.
> We hope the following answers your questions and addresses your concerns.
>
> > Q1. L 84: no extra cost?
> >
> > Q2. L 244: utilize/an assignment?
> >
> > Q3. Algorithm 1 L15: Is eq. (2) correctly eq. (3)?
>
> We are grateful to you for pointing these typos.
> We have corrected them in the *Rebuttal Revision* and highlighted the alterations in blue.
> If it is convenient, you can download and read the latest version.
>
> > Q4. The performances of CDS (Li et al. 2021, NeurIPS) in Figure 3 may be worse than expected (in the paper, CDS outperformed QMIX and QPLEX). I previously ran their code and it worked well. I know the experimental condition was different, but I want to know the reason (were there no code and hyperparameters for CDS?).
>
> Actually, we ran the experiments of CDS [1] using their provided [code](https://github.com/lich14/CDS).
> And due to the missing of a concrete document describing the choosing of parameters, the performance of CDS on the map 'academy_3_vs_1_with_keeper' in GRF is produced by the default parameters (file at *CDS/CDS_GRF/config/algs/CDS_QMIX.yaml* provided by CDS).
>
> There are some differences between our experiments and those in CDS that we believe may account for the unexpected results.
>
> First, as we described in Sec. 5.2, we add the randomness into the initialization while the environment in CDS initializes the agents in the fixed positions.
> Since CDS encourages the agents with different IDs to be diverse by maximizing the mutual information between the individual trajectory and agents' identity, the random initialization may be harmful to the performance (one extreme case: the agents exchange there positions but the IDs are the same).
>
> Second, the experiments of QMIX [2] in our work is produced by the default parameters at [pymarl](https://github.com/oxwhirl/pymarl/blob/master/src/config/default.yaml) where the parameter `obs_agent_id=True`, while the same parameter is set to be `False` in the default parameters provided by CDS. This parameter controls the inputs of the RNN and the agent's one-hot ID will be included in the observation when `obs_agent_id=True`, which means the network will not recognize different agents when it is set to be `False`.
>
> Besides, we find the experiments (Figure 3, 4) in [3] have similar results that CDS does not outperform QMIX and QPLEX [4].
>
> > Q5. There was no video at https://sites.google.com/view/spd-umarl.
>
> We are sincerely apologetic for releasing a wrong version at that time and we have updated the released site, please check our [site](https://sites.google.com/view/spd-umarl) again.
>
> > Q6. L328-329: “To avoid the effect of randomness, we repeat the visualization for 10 times for each skill.” It was not clear to me. What did the authors do?
>
> We are sorry for the confusion caused by this statement.
> To our concern that the different performance of different joint policies learned by SPD are only due to the randomness of the GRF environment itself but not the inter-policy diversity we want, we deploy each learned joint policy for 10 episodes and visualize them.
> The results show that each joint policy exhibits the similar coordination behavior across 10 episodes (as the videos at our [site](https://sites.google.com/view/spd-umarl) show).
>
> > Q7. Appendix L83: The results show that none of the URL approaches, including our method SPD, achieve more than 75% on DSR. The values in Fig. 2 a seem to be 60%. Why did the authors mention 75%?
>
> In fact, we evaluated the URL approaches with 5 different random seeds and the original results of these methods on DSR range from 56.6%~60.9%.
> Since there exist randomness during the evaluation process, we assume the result on DSR may surpass the current highest value.
> Actually, this sentence means that the performance of these URL approaches are far away from satisfying, implying that there is still space for improvement.
> We update the narration in the *Rebuttal Revision* and feel sorry for this confusion.
>
> [1] Chenghao, L., Wang, T., Wu, C., Zhao, Q., Yang, J., & Zhang, C. (2021). Celebrating diversity in shared multi-agent reinforcement learning. Advances in Neural Information Processing Systems, 34, 3991-4002.
>
> [2] Rashid, T., Samvelyan, M., Schroeder, C., Farquhar, G., Foerster, J., & Whiteson, S. (2018, July). Qmix: Monotonic value function factorisation for deep multi-agent reinforcement learning. In International conference on machine learning (pp. 4295-4304). PMLR.
>
> [3] Wu, S., Wang, T., Li, C., & Zhang, C. (2021). Containerized Distributed Value-Based Multi-Agent Reinforcement Learning. arXiv preprint arXiv:2110.08169.
>
> [4] Wang, J., Ren, Z., Liu, T., Yu, Y., & Zhang, C. (2020). Qplex: Duplex dueling multi-agent q-learning. arXiv preprint arXiv:2008.01062

---

> > ### Comment · Reviewer_meoo · 2022-08-09
> > **Thanks for the rebuttal**
> >
> > Thank you for the rebuttal.
> > My unclear points are clarified.
> > Although I have less confidence in my understanding, I raised my score.

---

### Official Review · Reviewer_tpzu · 2022-07-10

**Rating:** 7
**Confidence:** 4
**Soundness:** 3 good
**Presentation:** 3 good
**Contribution:** 4 excellent

**Summary:**


This work proposes a novel unsupervised reward labeling method called SPD, which rewards a population of agents when they have a relationship graph (synergy patten graph) that different to other populations. The

Concretely, this work first builds "synergy pattern graph" of a given population and then uses the SPG discrepancy between populations as the metric to encourage inter-agent diversity.

Experiments show that the proposed method can (1) learn diverse synergy patterns and (2) improve the generalizability on downstream tasks.

Generally, I like this work. I will raise my score if the writing can be improved and more convincing evaluation results, both qualitatively and quantitatively, can be provided.


# Update after responses

I am happy to see the authors revise the paper to make it more clear. I raised my score.


**Questions:**


### Questions on experiments


I saw StartCraft2 SMAC environment is supported in your code. Why don't put its result in the paper?

In Figure 2 (a), the d_sp is computed based on the Gromov-Wasserstein discrepancy between different populations. How you compute this for DIAYN and WURL? IIUC they only has one population. I think this figure is not convincing to stand for the claim "SDP captures the relative relations of agents" (line 338) and "SPD can inspire agents to visit identifiable states while further encouraging agents to explore with other types of coordination policies" (line 291). I expect more convincing evidence to show that after SPD pre-training the models already "learns useful synergy patterns" (line 338).

Can you plot the test performance of SPD population while conducting the unsupervised learning? If we can see performance improvement then it shows that SPD can learn policy without reward at all!






### Questions on method

If I understand correctly, there exists Z populations of policies and the population that actually involve in each episode is uniformly sampled from the pool of Z populations. Correct?
**Is it possible that this is the reason why SPD is working? This is similar to league training and ensemble method, which have shown can improve generalization.**
This is an important question. Please point out existing exp result in the paper if I miss it.

A question related to this is that do the Z populations really diverse? Figure 2 (a) shows the d_sp


---

Current method labels transitions with pseudo reward and stores them into replay buffer for QMIX training. Therefore the reward might be outdated in the process of training. Is it necessary to relabel those reward based on latest policies?


---


Related to above question, is it possible to compute the proposed pseudo reward based on truncated episode? For example, to generalize your method to on-policy RL algorithm like independent PPO, the reward should be computed on-the-fly before the episode is terminated.



### Other questions


Why the webpage is empty? If you don't prepare it, don't post it.

---

Some typos:

* CSD paper has wrong citation format.
* Algorithm 1 Line 14: Update d_gw according to Eq (6) as well as Eq (1).
* Line 139: "about how to [solving] the optimization problem"
* Line 244: "which could solve such [a] assignment"
* Line 288: can add a reference to d_sp


---

I don't think Figure 1 is informative.

---

Figure 2 (a) and (b) are in quite different styles. And the legend of Fig2(b) is sketchy. This could be improved.

---

Appendix C figure 1 is good, but can be improved. Agents in different teams should have clear difference. For example, opponent, ball and guard should use different shape rather than circle. I actually prefer to watch video, but unfortunately you don't provide video nor webpage.

---

Proposition 4.1 is boring. Maybe we can move it to appendix.


**Limitations:**



In Appendix D, the authors describe the limitations.

I think a limitation is that, supposing URL methods (including SPD) perform bad when simply deploying the learned models to the test tasks without further fine-tuning, how to make URL method works in a complete reward-free setting. Previous works in single-agent RL, like DIAYN, shows that we can learn policy without any reward. This also relates to the next limitation.

The second limitation is that, I think we can incorporate diversity encouraging technique in single-agent with the SPD. By doing this, we can improve the intra-agent diversity as well as inter-agent diversity.





**Strengths And Weaknesses:**


### Strengths

* This work formulates the diversity between agents in the perspective of "synergy pattern graph". This formulation of inter-agent diversity is novel. It encourages the **diversity in relationship**, which stands higher than simply computing diversity based on agents within one group or computing diversity based on single-agent information.
* The continuity and completeness of the writing is good. There are no obvious flaws and disconnected notations.
* I am very happy to see that code is provided. The code is provided with detailed documents and function description. This is a very big plus.


### Weaknesses

* The evaluation results do not provide a full view on the effectiveness of the proposed method. See questions.
* Writing clarity can be further improved since the notations are a bit overwhelmed. I think the terms like SPG, SPG batch, SPG element, Discrepancy of SP and so are messed. It would be better to simplify them if don't have enough space to clarify define and introduce them. Also, I expect to see a concrete form of reward in implementation section, rather than a notation that refers to another notation that refers to third notation.
* Besides, the connection between paragraphs and sections is not smooth enough, which creates huge burden in understanding. A good idea is to draw an illustrative diagram showing the relationship between concepts (I don't think Figure 1 is informative.).

---

> ### Author Response · Authors · 2022-08-02
> **Response to Reviewer tpzu (part 3)**
>
> > Q14. Appendix C figure 1 is good, but can be improved. ... nor webpage.
> >
> > Q15. Proposition 4.1 is boring. Maybe we can move it to appendix.
>
> Thanks for the advice and we've now improved the Appendix C Fig. 1 and moved Proposition 4.1 to Appendix A in the *Rebuttal Revision*.
>
> > Q16. In Appendix D, the authors describe the limitations ... inter-agent diversity.
>
> In fact, SPD does learn meaningful synergy patterns such as 'conducting ball in long distance' and 'passing-and-shooting' with no external reward from the environment and no further training (in Sec. 5.2 and Fig. 4), which is similar to the original single-agent experiments in DIAYN [3].
>
> We agree with the second limitation you mentioned and we've added it into the Appendix D.
>
> Finally, we are really grateful for the reviewer's detailedly review and suggestions.
> Besides, our work, to our knowledge, is the only method so far to use URL manner in the multi-agent settings and thus we also plan to open our source code for further studies.

---

> ### Author Response · Authors · 2022-08-02
> **Response to Reviewer tpzu (part 2)**
>
> > Q4. I expect more convincing evidence to show that after SPD pre-training the models already "learns useful synergy patterns" (line 338).
>
> In fact, as for the statement that SPD "learns useful synergy patterns", the experiments in Sec. 5.2 on GRF show that SPD can perform meaningful synergy patterns such as 'conducting ball in long distance' and 'passing-and-shooting' with no external reward from the environment and no further training, which is similar to the original single-agent experiments in DIAYN [3].
> Also, using the learned joint policies as the initialization in the downstream tasks can accelerate the learning preodure (as shown in Fig. 3) suggests some of the policies are already suitable for these specific tasks.
>
> [3] Eysenbach, B., Gupta, A., Ibarz, J., & Levine, S. (2018). Diversity is all you need: Learning skills without a reward function. arXiv preprint arXiv:1802.06070.
>
> > Q5. Can you plot the test performance of SPD population ... at all!
>
> Actually, we can not find significant trends at the learning curve of the environment returns during the URL learning process, which is acceptable since the agents never receive this signal at all.
> To this concern, we compare the test performance of the joint policies learned by different URL approaches with no extra training on the 'Half Court;Fixed Init' map as we used in Sec. 5.2 and that of a randomly initialized QMIX, and the results shows the chosen policy learned by SPD already exhibits some 'talent' on the downstream task while there is no task-related signal.
>
> |  Algorithm        | Episode Return  |
> |  ----             | ----  |
> | SPD               | $5.73 \pm 1.65$  |
> | DIAYN             | $0.17 \pm 0.24$ |
> | WURL              | $2.83 \pm 0.66$ |
> | QMIX              | $0.44 \pm 0.72$ |
>
> > Q6. If I understand correctly, there exists Z populations of policies ... in the paper if I miss it.
>
> Sorry for the confusion.
> As we have distinguished the difference between 'population' and 'synergy patterns' in the former part, here the process you concerned about actually is that, there exists Z policies and Z replay buffers and each policy samples one episode data and then insert it into **its own replay buffer** for training.
> During the downstream learning, we test Z policies on the task before training and choose the one with the best performance as the initialization for QMIX training.
> Therefore, it is quite different from 'league training' which adds copies of policies to the league and tries to against these past policies, in addition, 'ensemble method' which shares all data across the policies and has a set of policies during the whole learning procedure.
>
> > Q7. A question related to this is that do the Z populations really diverse? Figure 2 (a) shows the d_sp
>
> The $d_{sp}$ score in the Fig. 2(a) is the average of all the learned policies, which means each policy has enouge difference to the others.
> Besides, the visualization of GRF in Appendix C demonstrates that these policies do exhibit diversity.
>
> > Q8. Current method labels transitions with pseudo reward ... latest policies?
>
> Though the SPD computes the pseudo reward after the episode is terminated, the reward is amortized into each step, which matching the original training process of QMIX.
> Besides, the replay buffer for QMIX has a limits of 5000 samples and the oldest samples will be discarded, alleviating this problem during the training process.
>
> > Q9. Related to above question, is it possible to compute ... episode is terminated.
>
> 'No match', the variant of SPD discussed in the ablation part (in Sec. 5.1 and Appendix B.1), is exactly what you talk about.
> We set the matching function $\rho(t)=t$ directly which means there is no need to compute the pseudo-reward after the episode is terminated.
> And the result in Fig. 2(b) (Fig. 2 in the *Rebuttal Revision* now since we change the Fig. 2(a) into a table) shows the original SPD outperforms this variant.
>
> > Q10. Why the webpage is empty? If you don't prepare it, don't post it.
>
> We are really apologetic for releasing a wrong version at that time and we have updated the released site, please check our [site](https://sites.google.com/view/spd-umarl) again.
>
> > Q11. Some typos ...
>
> We are sincerely grateful for pointing these typos.
> We have corrected them in the *Rebuttal Revision* and highlighted the alterations in blue.
>
> > Q12. I don't think Figure 1 is informative.
>
> Fig. 1 shows that the pair of synergy pattern graphs are from step pair $(t, \rho^*(t))$ and we hope it helps the readers who are not familiar with the background to understand the process correctly.
> Thus, we insist to keep Fig. 1 since it delineates the process of obtaining the pseudo-reward for step $t$.
>
> > Q13. Figure 2 (a) and (b) are in quite different styles. And the legend of Fig2(b) is sketchy. This could be improved.
>
> Thank you for pointing this out.
> We have changed the Fig. 2(a) into a table and updated the legend of Fig. 2(b) in the *Rebuttal Revision*.

---

> > ### Comment · Reviewer_tpzu · 2022-08-04
> > **Figure 1 needs to be improved**
> >
> > Figure 1 is not a well-designed illustrative figure. I am not asking for removing it, but instead I will be happy to see it can be improved with less math notations, more diagram and serve as a good illustration figure.

---

> > > ### Author Response · Authors · 2022-08-09
> > > **Re: Figure 1 needs to be improved**
> > >
> > > Thank you so much for your advice.
> > >
> > > We've updated the Fig.1 in the revised submission and tried our best to illustrate the procedure vividly.
> > > However, we still keep a few necessary math notations to aid understanding.
> > >
> > > Hope this will alleviate your concern.

---

> ### Author Response · Authors · 2022-08-02
> **Response to Reviewer tpzu (part 1)**
>
> We sincerely appreciate your inspiring comments and insightful suggestions.
> We hope the following answers your questions and addresses your concerns.
>
> > Weakness. 2: Writing clarity can be further improved ... to third notation.
>
> Thanks for the advice.
> We have added a discrimination in Appendix F to clarify these concepts and a new equation Eq. (7) in Sec. 4.4 to show the final formulation of the pseduo-reward in the *Rebuttal Revision*, please download it if convenient.
>
> > Weakness. 3: Besides, the connection between paragraphs ... relationship between concepts.
>
> We are sorry for the non-smooth reading experience.
> We have been trying our best to smooth the flow between paragraphs in the *Rebuttal Revision* and highlighted the alterations in blue.
>
> > Q1. I saw StartCraft2 SMAC environment is supported in your code. Why don't put its result in the paper?
>
> We were struggled to fulfill the experiments on the different maps in GRF, and we were unable to evaluate the algorithms on SMAC due to the limitations of the computation resource.
> To the concern of the performance on SMAC, we will try our best to plot the further results before the end of 'Reviewer-Author Discussions' period.
>
> > Q2. In Figure 2 (a), the d_sp is computed ... one population.
>
> Firstly, we want to clarify the difference between the **'population'** you mentioned and the **'synergy pattern'** we proposed.
> IIUC, the 'population' you refer to is a collection of diverse policies for the agents to complete *a single task* through more efficient exploration (like [1]).
> Briefly, these learned policies are used for maximizing a *task-related* reward while keeping to be different to each other for exploration.
> URL (including our method SPD), however, aims to learn diverse skills/synergy patterns in *task-agnostic* settings, which means the learned policies are essentially different to the others.
> For instance, the learned synergy pattern 'pass-and-shoot' tends to be useful in the offensive tasks and the synergy pattern 'running back to play defense' is intuitively effective in the defensive tasks while they are both learned in one URL training process.
>
> As a matter of fact, DIAYN and WURL have the same number of policies to learn as our method SPD ($Z=10$ in the MPE experiment and $Z=20$ in the GRF experiment).
> Since they are designed for the single agent case, we deploy them by regarding all agents as a single agent.
> Concretely, we use the observations from all agents to create the features that DIAYN and WURL need.
> We have updated the description of these two baselines in the *Rebuttal Revision* to make a more clear delineation.
>
> [1] Parker-Holder, J., Pacchiano, A., Choromanski, K. M., & Roberts, S. J. (2020). Effective diversity in population based reinforcement learning. Advances in Neural Information Processing Systems, 33, 18050-18062.
>
> > Q3. I think this figure is not convincing ... coordination policies" (line 291).
>
> As we mentioned (line 287-288), we adopt $d_{sp}$ as the metric for measuring the discrepancy among relative relationships of agents since the SPG is built based on the relative relations.
> Besides, to our best knowledge, there is no other suitable metric for the relative relations of agents and that's also why we proposed the discrepancy of synergy patterns.
> As for the effectiveness of $d_{sp}$, we follow the suggestions by *Reviewer 8Kca* and *Reviewer tFW2*, adding further evaluation of DIAYN and WURL on GRF.
> We plot the learning curves of DIYAN and WURL in Fig. 3 in the *Rebuttal Revision*.
> The results shows that SPD learns faster and finally gets higher winning rate on all scenarios compared to the baseline QMIX and single-agent URL approaches.
> We believe the gap between the improved performances of SPD and the other URL approaches is caused by the fact that SPD does capture more information than the single-agent URL approaches in multi-agent settings, which is the relative relations of agents since it is designed for this.
>
> As for the claim that "SPD can inspire agents to visit identifiable states while further encouraging agents to explore with other types of coordination policies" (line 291), we have updated it to "SPD can inspire agents to visit **comparably** identifiable states **as the conventional URL approaches** while further encouraging agents to explore with other types of coordination policies".
> The results on the DSR which measures the diversity of states shows that SPD achieves similar performance as DIAYN and WURL, and SPD significantly outperms conventional URL approaches on $d_{sp}$ which measures the discrepancy among relative relationships of agents.
> By "... explore with other types of coordination policies", we mean the learned joint policies are diverse.

---

### Official Review · Reviewer_tFW2 · 2022-07-10

**Rating:** 4
**Confidence:** 4
**Soundness:** 1 poor
**Presentation:** 2 fair
**Contribution:** 2 fair

**Summary:**

This paper proposes an unsupervised framework SPD to discover diverse policies for agents with no extrinsic reward. SPD uses synergy pattern graph to represent the joint policy for agents and discovers diversity policy by applying optimal transport. The experiments show that SPD outperforms some MARL algorithms in some environments. The contributions of this work are designing a new method to represent the joint policy for agents by considering other agents in MARL and proposing to explore skills by optimal transport.

**Questions:**


Questions:
1. Why is CDS performance worse than QMIX?

2. Why not compare with some other methods of URL on google football? Although these URL methods are not designed for the MARL algorithms, they are needed for comparison.



Suggestions:

I think the auther should discuss some works that are designed to encourage diversity [1] [2] [3].
Furthermore, I believe the baselines that are compared in this work are weak. Specifically, some MA exploration works are not considered, and the baseline with the best performance in the experiments (i.e., football game) is QMIX. The authors need to consider some effective MA exploration works [1].
In addition, the experiments do not compare URL methods on google football and more URL methods should be taken into consideration, e.g., [4] [5].
Finally, SPD is compared on some environments and more environments need to be considered, such as SMAC, MAMUJOCO.
For example, on some SMAC or on Google football environments used by CDS. Thus a more comprehensive experiments could significantly improve the paper.


[1]: Zhou Z, Fu W, Zhang B, et al. Continuously Discovering Novel Strategies via Reward-Switching Policy Optimization[J]. arXiv preprint arXiv:2204.02246, 2022.

[2]: Parker-Holder J, Pacchiano A, Choromanski K M, et al. Effective diversity in population based reinforcement learning[J]. Advances in Neural Information Processing Systems, 2020, 33: 18050-18062.

[3]: Lupu A, Cui B, Hu H, et al. Trajectory diversity for zero-shot coordination[C]//International Conference on Machine Learning. PMLR, 2021: 7204-7213.

[4]: Lee L, Eysenbach B, Parisotto E, et al. Efficient exploration via state marginal matching[J]. arXiv preprint arXiv:1906.05274, 2019.

[5]: Liu H, Abbeel P. Aps: Active pretraining with successor features[C]//International Conference on Machine Learning. PMLR, 2021: 6736-6747.

**Ethics Review Area:**

["I don’t know"]

**Limitations:**

Limitations should be discussed more adequately in the main body of the paper. Societal impact is likely minimal.

**Strengths And Weaknesses:**

Strengths:

1. This work provides an interesting and novel idea on the use of graphs to represent the joint policy for MARL.

2. The paper is well written.

Weaknesses:

1. There is no support for the conclusion that SPD is better than other URL methods in terms of performance.

2. Some standard benchmarks used in the MARL community need to be considered. Some of the results are very confusing to me, such as CDS does not work at all.

---

> ### Author Response · Authors · 2022-08-02
> **Response to Reviewer tFW2**
>
> Thank you for your feedback and suggestions. We hope the following answers your questions and addresses your concerns.
>
> > Q1. Why is CDS performance worse than QMIX?
>
> We ran the experiments of CDS using their provided [code](https://github.com/lich14/CDS) with the default parameters (file at *CDS/CDS_GRF/config/algs/CDS_QMIX.yaml* provided by CDS).
>
> In fact, there are some differences between our experiments and those in CDS that we believe may account for the worse performance of CDS.
> First, as we described in Sec. 5.2, the randomness is added into the initialization while the environment in CDS initializes the agents in the fixed positions.
> This random initialization may damage the performance of CDS because CDS inspires the agents with different IDs to be diverse by maximizing the mutual information between the individual trajectory and agents' identity. (one extreme case: the agents exchange there positions but the IDs are the same).
> Second, the experiments of QMIX in our work is produced by default parameters at [pymarl](https://github.com/oxwhirl/pymarl/blob/master/src/config/default.yaml) where the parameter `obs_agent_id=True`, while the same parameter is set to be `False` in the default parameters provided by CDS.
> The inputs of the RNN is controlled by this parameter and the agent's one-hot ID will be included in the observation when `obs_agent_id=True`, which means the network will not recognize different agents when it is set to be `False`.
>
> Besides, the experiments (Figure 3, 4) in [6] have similar results that CDS mostly does not outperform QMIX and QPLEX.
>
> [6] Wu, S., Wang, T., Li, C., & Zhang, C. (2021). Containerized Distributed Value-Based Multi-Agent Reinforcement Learning. arXiv preprint arXiv:2110.08169.
>
> > Q2. Why not compare with some other methods ... for comparison.
>
> Thanks for your important advice.
> We've compared with DIAYN [5] and WURL [6] on the GRF experiment with the same setting as SPD in Sec. 5.2 in the *Rebuttal Revision*, please check it if convenient.
> The results show that using the model learned by conventional URL methods (WURL and DIAYN) as the initialization mostly performs similarly to the baseline QMIX, while WURL slightly outperforms QMIX on the 'Full-court' maps.
> In contrast, SPD learns faster and finally gets higher winning rate on all scenarios.
> As for the performance of WURL on the 'Full-court' maps which need more exploration, the models trained by WURL may reach diverse states during the URL training and we believe this accounts for the slightly better performance.
> The significant improvement of SPD demonstrate that SPD is better for multi-agent settings compared to these URL approaches.
>
> > Suggestions 1. I think the auther should discuss some works ... MA exploration works [1].
>
> We would like to point out the difference between unsupervised skill discovery and these diversity-based exploration methods([1, 2, 3]).
> Unsupervised Reinforcement Learning (URL), including our method SPD, mainly aims to learn diverse skills/synergy patterns in task-agnostic settings.
> The diversity that URL cares about is the inter-policy diversity.
> And the results in Sec. 5.2 and Appendix C show that SPD does learn useful synergy patterns with no external reward from the environment.
> The downstream task (in our paper, the experiment on GRF) learning is one of the applications of the policies learned by URL.
> Meanwhile, these diversity-based exploration methods mostly enhance single policy’s diversity or the inter-agent diversity and aim to encourage the exploration for efficient learning.
>
> Moreover, the reason why we compared with CDS is that it was reported to be the SOTA on GRF and we did not find the experiment on GRF in these methods, while we did not find official code provided by them as well.
> However, these diversity-based methods are related and we have cited and discussed them in the Related Works part in the *Rebuttal Revision*.
> Thank you for pointing out this.
>
> > Suggestions 2.  In addition, the experiments do not ... consideration, e.g., [4] [5].
>
> Actually, Lee et al. [4] is also a exploration method that utils the intrinsic reward for exploration, which aims to learn **a single policy** for which the state marginal distribution matches a given target state distribution.
>
> As for Liu et al. [5], we've been working on implementing this algorithm and we will try our best to plot the further comparison result before the deadline of 'Reviewer-Author Discussions' period.
>
> > Suggestions 3. Finally, SPD is compared on some environments ...  improve the paper.
>
> Our experiment on 'academy_3_vs_1_with_keeper' in GRF is actually the same map used by CDS except we add randomness and increase the difficulty of exploration.
> To the conerned point, we are working on produce further results on SMAC.
> We believe these further evaluation will strengthen our claim on superiority of SPD.
>
> > limitations
>
> We have discussed limitations in the Appendix D due to the page limits by NeurIPS.

---

### Author Response · Authors · 2022-08-09
**General Response to all Reviewers**

We sincerely appreciate all reviewers for their time and efforts in evaluating our paper, as well as their detailed comments and suggestions.

We hope that our responses and answers to your questions will alleviate your concerns and further improve your opinion of our work.
A revised version of our paper and supplementary material has been uploaded.
We would be pleased to answer any further questions you may have.

---

### Meta-Review · Area_Chair_26aX · 2022-08-26

**Recommendation:** Accept
**Confidence:** Less certain

**Metareview:**

Reviewers appreciated the paper's contribution of a novel method for unsupervised skill learning in MARL. While the scores were borderline, reviewers are mostly in favor of acceptance, therefore I recommend acceptance as well. Additional baselines and environments added during the rebuttal phase were important considerations in this decision.

**Award:**

No

---

### Decision · Program_Chairs · 2022-09-14

Accept